# A SPLICS reporter reveals $\alpha$-synuclein regulation of lysosome-mitochondria contacts which affects TFEB nuclear translocation

Flavia Giamogante[1,10], Lucia Barazzuol[1,10], Francesca Maiorca [2], Elena Poggio[2], Alessandra Esposito [3,4], Anna Masato[2,5], Gennaro Napolitano [3,4], Alessio Vagnoni [6], Tito Calì [1,7,8] ✉ & Marisa Brini [2,8,9] ✉

Mitochondrial and lysosomal activities are crucial to maintain cellular homeostasis: optimal coordination is achieved at their membrane contact sites where distinct protein machineries regulate organelle network dynamics, ions and metabolites exchange. Here we describe a genetically encoded SPLICS reporter for short- and long- juxtapositions between mitochondria and lysosomes. We report the existence of narrow and wide lysosome-mitochondria contacts differently modulated by mitophagy, autophagy and genetic manipulation of tethering factors. The overexpression of $\alpha$-synuclein ($\alpha$-syn) reduces the apposition of mitochondria/lysosomes membranes and affects their privileged $Ca^{2+}$ transfer, impinging on TFEB nuclear translocation. We observe enhanced TFEB nuclear translocation in $\alpha$-syn-overexpressing cells. We propose that α-syn, by interfering with mitochondria/lysosomes tethering impacts on local $Ca^{2+}$ regulated pathways, among which TFEB mediated signaling, and in turn mitochondrial and lysosomal function. Defects in mitochondria and lysosome represent a common hallmark of neurodegenerative diseases: targeting their communication could open therapeutic avenues.

Organelle contact sites are increasingly recognized as the bottleneck where nutrients, metabolites, ions, and lipids are fine-tuned to maintain cell homeostasis[1]. Indeed, even subtle variations in the bidirectional communication between different organelles can have a big impact on the biophysical and functional characteristics of the organelles themselves, and overall on the cell physiology, resulting in early sensitization to cell dyshomeostasis. Organelles tethering is hijacked in many pathological conditions and its molecular characterization has recently attracted major interest since it may represent a target for therapeutic intervention[2–6]. In particular, mitochondria and lysosomes are functionally[7] and physically[8–11] strictly interconnected being their dysfunction shared by many neurological disorders[12,13]. Despite the possibility that their crosstalk could be an important hub for neuronal homeostasis is recently emerging[2,9,14–17], the study of their interaction

[1]Department of Biomedical Sciences (DSB), University of Padova, Padova, Italy. [2]Department of Biology (DIBIO), University of Padova, Padova, Italy. [3]Telethon Institute of Genetics and Medicine (TIGEM), Naples, Italy. [4]Department of Medical and Translational Science, Federico II University, Naples, Italy. [5]UK-Dementia Research Institute at UCL, University College London, London, UK. [6]Department of Basic and Clinical Neurosciences, Maurice Wohl Clinical Neuroscience Institute, Institute of Psychiatry, Psychology and Neuroscience, King's College London, London, UK. [7]Padova Neuroscience Center (PNC), University of Padova, Padova, Italy. [8]Study Center for Neurodegeneration (CESNE), University of Padova, Padova, Italy. [9]Department of Pharmaceutical and Pharmacological Sciences (DSF), University of Padova, Padova, Italy. [10]These authors contributed equally: Flavia Giamogante, Lucia Barazzuol. ✉e-mail: tito.cali@unipd.it; marisa.brini@unipd.it

at specific membrane contact sites is still largely unexplored. Multiple studies using different imaging techniques have demonstrated that interorganelle contact sites between mitochondria and lysosomes occur at an average distance between membranes of ~10 nm[14] and can dynamically form under healthy conditions[18–20]. These contacts are clearly distinct from the contacts occurring under lysosomal degradation routes such as mitophagy[21] or involving mitochondrial-derived vesicles[22]. In this respect, molecular details at the basis of their dynamics have revealed the existence of a tight regulation by multiple proteins among which the small GTPase Rab7 is one of the major players. In its GTP-bound state it localizes to lysosomes and, by binding possible effector proteins on mitochondria, acts as a master regulator promoting contact formation. Rab7 GTP hydrolysis to a GDP-bound state mediates lysosome-mitochondria contact untethering[9] and is driven by the Rab7 GTPase activating protein (GAP) TBC1D15, bound to the outer mitochondrial membrane protein Fis1[23,24]. The inability to undergo GTP hydrolysis, as observed upon the expression of the constitutively active Rab7 (Q67L)-GTP mutant, increased the number of contacts and prolonged the duration of the tethering. Accordingly, the inhibition of Rab7 GTP hydrolysis by the TBC1D15 (D397A) GAP-domain mutant led to inefficient untethering[9]. Functionally, lysosome-mitochondria contacts play key roles in regulating the dynamics of both mitochondria[8,9,25,26] and lysosomes[8,27] as well as metabolite homeostasis through bidirectional exchange of Ca²⁺, cholesterol, and iron[28–32]. Recently, impaired formation and function of lysosome-mitochondria contact sites has been associated with the onset of different neurodegenerative diseases i.e., Charcot-Marie-Tooth (CMT) disease[20,33]; Mucolipidosis type IV (MLIV)[29], Niemann-Pick type C (NPC)[30] and Parkinson's disease (PD)[18,34], suggesting that their modulation might represent a potential mechanistic pathway contributing to neurodegeneration, and thus become a target of possible therapeutic intervention. However, the molecular details of the consequences of impaired lysosome-mitochondria contact in PD pathogenesis are currently unknown. The gap is further hampered by the lack of reporters to easily image lysosome-mitochondria proximity over a range of distances in living cells and in vivo. We had previously designed split-GFP based contact site sensors of organelle proximity and tested them in vitro and in vivo[35–38].

Here we report on the generation of a reporter for lysosome-mitochondria proximity (SPLICS$_{S/L}$-P2A$^{LY-MT}$) and on its validation in conditions well known to impact on lysosome-mitochondria tethering. We describe the existence of at least two types of lysosome-mitochondria contact sites in human cells, narrow and wide, which differently respond to mitophagy or autophagy stimuli and to changes in the levels of specific tethering/untethering factors. Here we also report on the application of the SPLICS$_{S/L}$-P2A$^{LY-MT}$ reporter to explore lysosome-mitochondria contact sites in model cells overexpressing the PD-related protein α-synuclein (α-syn). Mitochondria and lysosome dysfunction have been widely reported to contribute to the onset of many neurodegenerative conditions, including PD, and α-syn gene (SNCA) multiplication[39–42] or point mutations are causally linked to dominant familial early-onset parkinsonism. Since α-syn interferes with both mitochondria and lysosome activities[43–45], we have directly tested whether α-syn accumulation, i.e., one of the most pathological hallmark of PD[46], could modulate the lysosome-mitochondria interface. Furthermore, we have searched for possible consequences of α-syn overexpression on the pathway of transcription factor EB (TFEB), the master regulator of autophagy, lysosomal biogenesis, and in intracellular clearance pathways which has been widely proposed as therapeutic target in cellular and mouse models for lysosomal storage disorders[47–50] and neurodegenerative diseases[51–56]. We show here that the overexpression of WT or the PD-related A30P and A53T α-syn mutants strongly impaired lysosome-mitochondria tethering, impinging on the efficient mitochondrial Ca²⁺ uptake upon its release from the lysosomes, thus reducing mitochondrial Ca²⁺ buffering and

consistent with increasing local cytosolic Ca²⁺ concentration. Intriguingly, the cells overexpressing α-syn WT also display augmented TFEB nuclear localization, possibly by Ca²⁺-calcineurin activation and TFEB dephosphorylation[47,57–65]. In summary, we show here the development and characterization of a genetically encoded SPLICS reporter for in vitro and in vivo monitoring lysosome-mitochondria contact sites and propose that their modulation could open avenues to tune the TFEB pathway and the downstream protective cell responses, i.e., autophagy and lysosomal biogenesis, to counteract neurodegenerative processes.

## Results

### Generation and characterization of SPLICS$_{S/L}$-P2A$^{LY-MT}$ reporter for lysosomes mitochondria contact sites

By exploiting the plasticity of split-GFP and of the second SPLICS generation characterized by P2A peptide sequence ensuring the equimolar expression of organelle-targeted GFP fragments[35,36], we have generated a SPLICS sensor (SPLICS$_{S/L}$-P2A$^{LY-MT}$) to detect tethering between lysosomes and mitochondria (Fig. 1a). To achieve this goal, we have fused the β11 strand of the GFP to the C-terminus of the full-length human TMEM192 protein[66] to target it at the lysosomal membrane. TMEM192 exposed both N- and C- termini on the cytosolic face and it is widely employed in the generation of chimeric proteins for its high specificity, ability to retain lysosomal localization upon overexpression, lack of glycosylation and perturbations of the lysosomal and cell's physiology[67–70]. Two glycine-repeat linkers of different length were introduced between the TMEM192 and the β11 strand fused moiety: the short linker was designed to detect interactions occurring over a range of ≈4 nm (SPLICS$_S$), while the longer one to detect interactions at ≈10 nm (SPLICS$_L$) (Fig. 1b). The non-fluorescent GFP$_{1-10}$ moiety was targeted to the cytosolic face of the outer mitochondria membrane as previously described (OMM-GFP$_{1-10}$)[35,36]. To validate our probe HeLa cells were transfected with SPLICS$_{S/L}$-P2A$^{LY-MT}$ expressing vectors and observed under confocal microscope. An evident green fluorescence puncta pattern is appreciable from images in Fig. 1c, both for short- and long-range contacts, due to the reconstitution of the two split SPLICS$_{S/L}$-P2A$^{LY-MT}$ GFP moieties at lysosome-mitochondria (Ly-Mt) contact sites. Importantly the reconstituted GFP fluorescence colocalized with both the outer mitochondria membrane marker (Tom20) and lysosome-associated membrane marker (LAMP1), as revealed by the immunocytochemistry analysis (Fig. 1c). The signal at the interface between lysosomes and mitochondria has been quantified as previously described[37] and, interestingly, the number of Ly-Mt long-range contacts (Mean ± SEM: SPLICS$_L$-P2A$^{LY-MT}$ 94.7 ± 4.4 $n = 35$) was significantly higher respected to short-range ones (Mean ± SEM: SPLICS$_S$-P2A$^{LY-MT}$ 72 ± 4.4 $n = 37$; ***p ≤ 0.001) (Fig. 1d). The correct mitochondria targeting of the OMM-GFP$_{1-10}$ moieties generated by these two probes was further verified by immunocytochemistry analysis on SPLICS$_{S/L}$-P2A$^{LY-MT}$ transfected cells using an anti-GFP antibody which detects the GFP$_{1-10}$ portion and whose signal perfectly colocalized with the signal detected by an anti-Tom 20 antibody (Fig. 1e). The correct targeting of the β11 portion driven by TMEM192 was verified by the generation of a Ly-GFP$_{1-10}$ construct, where the GFP$_{1-10}$ portion was fused at the C-terminus of TMEM192. When the Ly-GFP$_{1-10}$ construct was co-transfected with a Kate-β11 expressing vector[35], a typical lysosomal pattern was revealed after GFP reconstitution, which perfectly matched LAMP1 signal (Fig. 1f, top panels). Similarly, when the Ly-GFP$_{1-10}$ construct was transfected alone, an anti-GFP antibody stained the GFP$_{1-10}$ portion whose signal perfectly colocalized with the signal detected by an anti-LAMP1 antibody (Fig. 1f, bottom panels). The characterization of the probe has been completed by a Western blot analysis that confirmed equal levels of expression for both the TMEM192-β11 (Fig. 1g) and the OMM-GFP$_{1-10}$ (Fig. 1h) portions upon the transfection of both SPLICS$_{S/L}$-P2A$^{LY-MT}$ short and long constructs. The staining with anti-TMEM192 and anti-GFP primary

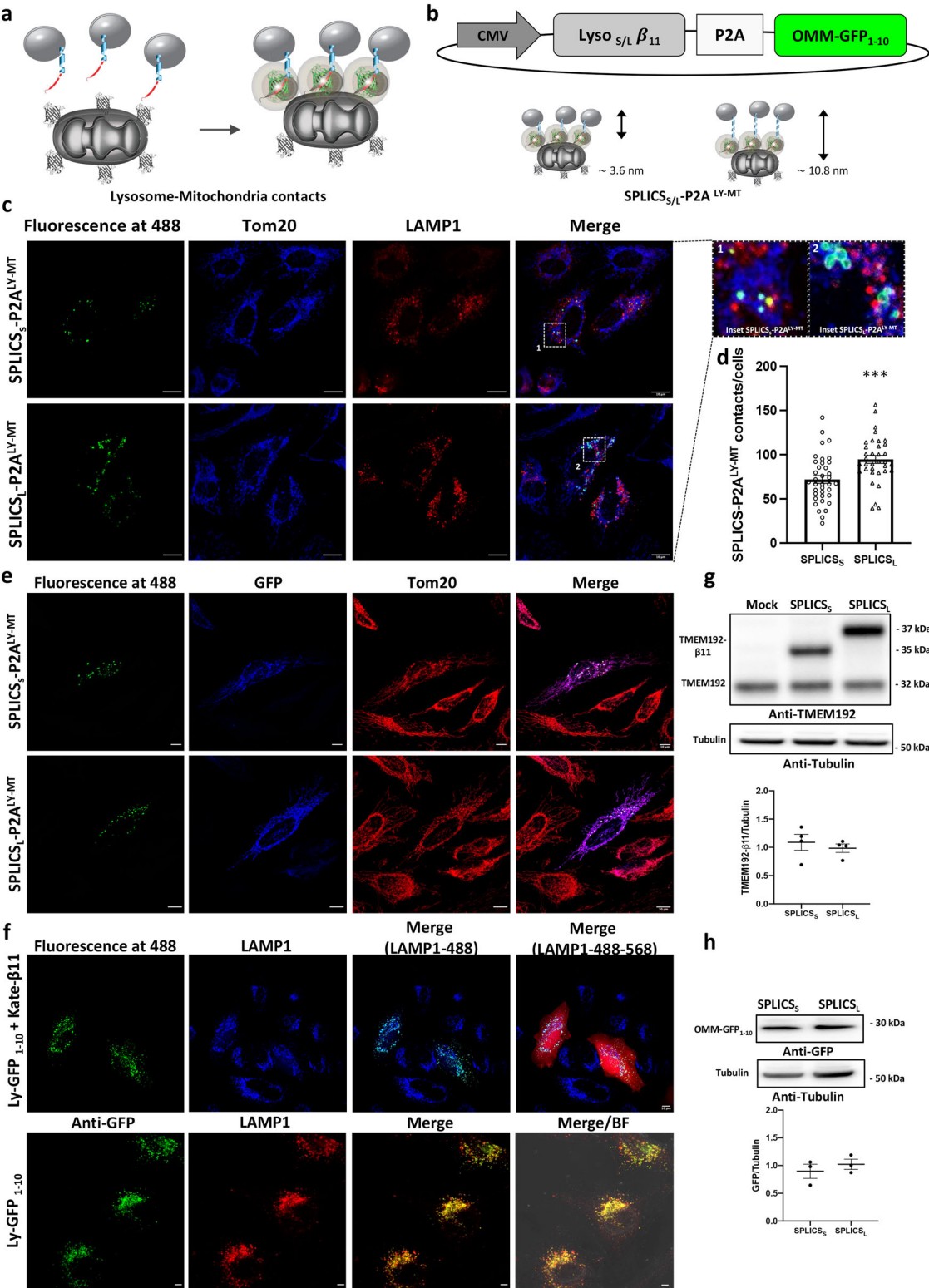

antibody, respectively, excluded the possibility that the differences in the number of contacts sites detected by the two SPLICS probes (short and long) could be dependent on differences in the protein amount of the two reporters.

## Short- and long-range lysosomes–mitochondria interactions are differentially modulated by Rab7A and TBC1D15

As a proof-of-concept experiment, the flexibility of the SPLICS$_{S/L}$-P2A$^{LY-MT}$, in terms of ability to detect subtle changes in the number of contact sites, has been tested by altering the expression of proteins involved in the formation of Ly-Mt contacts. The data available from literature[7] identified the GTP-binding protein Rab7A, enrolled on late endosomes/lysosomes, and the GTPase activating protein TBC1D15, recruited on the mitochondrial surface as the main proteins participating to Ly-Mt tethering and untethering processes (Fig. 2a)[8]. In this respect, we co-transfected HeLa cells with SPLICS$_{S/L}$-P2A$^{LY-MT}$ in presence of WT or mutated Rab7A or TBC1D15. Obtained data showed that, while the WT Rab7A affected in a significant manner the number

**Fig. 1 | Design and functional characterization of the SPLICS$_{S/L}$-P2A$^{LY-MT}$ probes.**
**a** Cartoon of the split GFP non-fluorescent portions, GFP$_{1-10}$ and β11, targeted to outer face of lysosomes and mitochondria. **b** Schematic representation of the SPLICS$_{S/L}$- P2A$^{LY-MT}$ vectors. The Lyβ11$_{S/L}$ coding sequence is cloned upstream the viral P2A peptide sequence, while the second cassette is occupied by the OMM-GFP$_{1-10}$. SPLICS$_{S/L}$-P2A$^{LY-MT}$ probes reveal contact sites occurring in the 4 nm to 11 nm range. **c** Representative Z-projection images of HeLa cells expressing the SPLICS$_{S/L}$- P2A$^{LY-MT}$ probes. Fluorescence at 488 refers to the complementation of GFP (488 nm) at LY-MT interface. Anti-Tom20 (mitochondria, blue) and anti-LAMP1 (lysosomes, red). Zoomed regions are shown in the top left panel. **d** Quantification of LY-MT short and long contacts in HeLa cells. The SPLICS dots were counted in the 3D rendering of a complete Z-stack, mean ± SEM: SPLICS$_L$-P2A$^{LY-MT}$ 94.7 ± 4.4 $n = 35$ and SPLICS$_S$-P2A$^{LY-MT}$ 72 ± 4.4 $n = 37$ cells examined over 3 independent experiments. **e** Representative Z-projection images of HeLa cells expressing the SPLICS$_{S/L}$- P2A$^{LY-MT}$ probes. Fluorescence at 488 refers to the complementation of GFP (488 nm) at LY-MT interface. Anti-GFP and anti-Tom20 were used to identify the OMM-GFP$_{1-10}$ (in blue) and mitochondria (in red), respectively. **f** Representative

Z-projection images of HeLa cells co-expressing LyGFP$_{1-10}$ and Kateβ11. Fluorescence at 488 refers to their complementation at lysosomes level. Anti-LAMP1 (lysosome, blue). The merges reveal either the co-localization between lysosomes and complemented LyGFP$_{1-10}$/Kateβ11(LAMP1-488) or between lysosomes, the complemented LyGFP$_{1-10}$/Kateβ11 and the cytosolic Kate β11 (LAMP1-488-568). In the bottom panel HeLa cells expressing the LyGFP$_{1-10}$ construct are shown, fluorescence at 488 refers to the anti-GFP immunofluorescence while anti-LAMP1 was used to identify lysosomes (red). The merge reveals the co-localization between LyGFP$_{1-10}$ and lysosomes with and without bright field. **g** Western blotting of HeLa cells expressing SPLICS$_S$- P2A$^{LY-MT}$ and SPLICS$_L$- P2A$^{LY-MT}$ reporters. Anti-TMEM192 antibody reveals the presence of endogenous (lower bands at 32 kDa) or transfected (upper bands at 35 and 37 kDa, respectively) TMEM192β11. **h** Western blotting with an anti-GFP antibody reveals the 30 kDa OMM-GFP$_{1-10}$ band. Quantitative analysis was performed on 4 (**g**) or 3 (**h**) independent experiments. Equal amount of total loaded lysate was verified by incubation with anti-β tubulin antibody. Scale bar 10 μm, (***$p \leq 0.001$ unpaired two-tailed $t$ test). Source data are provided as a Source Data file.

of short Ly-Mt contacts, the WT TBC1D15 did not alter this interaction (Mean ± SEM: SPLICS$_S$- P2A$^{LY-MT}$ + Mock 76.8 ± 3.5 $n = 60$ vs SPLICS$_S$-P2A$^{LY-MT}$ + Rab7A 95.9 ± 9.6 $n = 43$ *$p \leq 0.05$; SPLICS$_S$- P2A$^{LY-MT}$ + TBC1D15 63.2 ± 5.5 $n = 33$) (Fig. 2b, d). Interestingly, the number of long Ly-Mt contact sites was not affected by the overexpression of both Rab7A WT and TBC1D15 WT (Mean ± SEM: SPLICS$_L$- P2A$^{LY-MT}$ + Mock 97.4 ± 3.9 $n = 52$; SPLICS$_L$- P2A$^{LY-MT}$ + Rab7A WT 105.8 ± 6.5 $n = 30$ *$p \leq 0.05$; SPLICS$_L$- P2A$^{LY-MT}$ + TBC1D15 WT 81.9 ± 4.6 $n = 35$) (Fig. 2b, e). To better validate the SPLICS sensitivity to lysosome-mitochondria interplay alteration, the same analysis has been performed using two mutants as positive controls, the constitutively GTP-bound Rab7A (Q67L), and the constitutively inactive GTPase activating protein TBC1D15 (D397A), which have been previously reported to induce contact sites formation or their untethering, respectively (Fig. 2a). We also used a negative mutant, the constitutively GDP-binding Rab7A (T22N) that blocks the Ly-Mt contact sites formation (Fig. 2a). Notably, the quantification of SPLICS$_S$- P2A$^{LY-MT}$ signal has revealed that both Rab7A (Q67L) and TBC1D15 (D397A) overexpression strongly increased the mean of short contacts while, surprisingly, the negative mutant Rab7A (T22N) did not significantly change their number (Mean ± SEM: SPLICS$_S$- P2A$^{LY-MT}$ + Rab7A (Q67L) 123.8 ± 9.8 $n = 39$ **$p \leq 0.01$; SPLICS$_S$- P2A$^{LY-MT}$ + TBC1D15 (D397A) 103.2 ± 7.1 $n = 38$ **$p \leq 0.01$; SPLICS$_S$- P2A$^{LY-MT}$ + Rab7A (T22N) 71.9 ± 7.7 $n = 37$) (Fig. 2b, d). It was of interest that the expression of Rab7A (T22N) was the only condition able to significantly decreased the Ly-Mt long contacts compared to mock cells while both Rab7A (Q67L) and TBC1D15 (D397A) did not affect them (Mean ± SEM: SPLICS$_L$- P2A$^{LY-MT}$ + Rab7A (Q67L) 106.4 ± 6.8 $n = 31$; SPLICS$_L$-P2A$^{LY-MT}$ + TBC1D15 (D397A) 108.5 ± 7.2 $n = 37$; SPLICS$_L$- P2A$^{LY-MT}$ + Rab7A (T22N) 75 ± 5.8 $n = 34$ *$p \leq 0.05$) (Fig. 2b, e). No gross changes in the number and distribution of lysosomes (and mitochondria) were observed upon the overexpression of Rab7A and TBC1D15 variants as reported by LAMP1 or TOM20 immunofluorescence performed in cells co-expressing the SPLICS construct (Supplementary Fig. 1a, b) or expressing the Rab7A and TBC1D15 variants alone (Supplementary Fig. 1c). Fluorescence at 568 or 594 nm documented the presence of overexpressed Rab7A and TBC1D15 variants and Supplementary Fig. 1d confirmed that Rab7A and TBC1D15 variants were properly expressed. These data strengthened the reliability of SPLICS probes to detect changes in the number of Ly-Mt interactions occurring at different distances: short Ly-Mt contacts were finely sensitive to the overexpression of WT Rab7A, constitutivelt active Rab7A (Rab7A(Q67L)), and inactive TBC1D15 (TBC1D15(D397A)) while the long ones were more susceptible to the presence of inactive Rab7A (Rab7A (T22N)). Radial analysis was also performed to exclude that changes in the contacts number could be due to non-specific perinuclear clustering of lysosomes and/or membrane crowding induced by Rab7 (Supplementary Fig. 2 panels a and b).

## Effects of Rab7A and TBC1D15 silencing on short- and long-range lysosomes–mitochondria contact sites

To better evaluate the sensitivity of SPLICS$_{S/L}$- P2A$^{LY-MT}$ to detect changes in Ly-Mt interaction, we silenced Rab7A and TBC1D15 by using two different siRNA for each protein. Supplementary Fig. 1e, f documented that the silencing was effective for all the selected siRNAs. Interestingly, the silencing of Rab7A (Fig. 2c) did not affect short contact sites (Mean ± SEM: SPLICS$_S$- P2A$^{LY-MT}$ + SCR 79.5 ± 5.8 $n = 40$; SPLICS$_S$- P2A$^{LY-MT}$ + siRNA Rab7A #1 83.8 ± 8.1 $n = 33$; SPLICS$_S$- P2A$^{LY-MT}$ + siRNA Rab7A #2 79.4 ± 6.2 $n = 32$; Fig. 2f), but instead reduced the long ones, i.e., the siRNA Rab7A #1 significantly reduced the number of long Ly-Mt contacts while a downward trend appeared evident with the siRNA Rab7A #2 (Mean ± SEM: SPLICS$_L$- P2A$^{LY-MT}$ + SCR 107.1 ± 5.8 $n = 47$; SPLICS$_L$- P2A$^{LY-MT}$ + siRNA Rab7A #1 80 ± 5.2 $n = 39$ *$p \leq 0.05$; SPLICS$_L$- P2A$^{LY-MT}$ + siRNA Rab7A #2 91 ± 7.7 $n = 30$; Fig. 2g). Differently, the silencing of TBC1D15 significantly increased the number of short Ly-Mt interactions using both TBC1D15 siRNAs (Mean ± SEM: SPLICS$_S$-P2A$^{LY-MT}$ + siRNA TBC1D15 #1 110.2 ± 7.1 $n = 34$; SPLICS$_S$- P2A$^{LY-MT}$ + siRNA TBC1D15 #2 97.9 ± 6.3 $n = 40$ *$p \leq 0.05$, **$p \leq 0.01$) (Fig. 2c, f), but it was ineffective on the long ones ((Mean ± SEM: SPLICS$_L$- P2A$^{LY-MT}$ + siRNA TBC1D15 #1 95.3 ± 8 $n = 37$; SPLICS$_L$- P2A$^{LY-MT}$ + siRNA TBC1D15 #2 101.4 ± 9.1 $n = 34$)) (Fig. 2c, g). Overall, these results suggest that our SPLICS$_{S/L}$- P2A$^{LY-MT}$ probe is well suited to reveal differences in the formation of short and long Ly-Mt contact sites. An increase in Ly-Mt short contact sites was observed upon activation of Rab7A (or by overexpressing it or by inhibiting(TBC1D15)), and a reduction of Ly-Mt long contact sites was detected by reducing Rab7A activity (or by silencing it or by expressing a dominant negative form) suggesting the existence of a tunable regulation dependent on the degree/entity of Rab7A activity.

## Modulation of lysosome–mitochondria contact sites during autophagy and mitophagy

To challenge whether our SPLICS$_{S/L}$-P2A$^{LY-MT}$ probe was able to rule out some biological issues related to mitochondria and lysosome crosstalk under autophagy/mitophagy conditions, we carried out Ly-Mt contact sites quantification upon lysosomal and mitochondrial stressing conditions. HBSS and rapamycin application were used to induce cellular starvation condition (Supplementary Fig. 3) and Carbonyl Cyanide Chlorophenylhydrazone (CCCP) and oligomycin/antimycin A (OA/AA) treatment to evoke mitophagy. We found that only short Ly-Mt interactions were affected during autophagy and mitophagy activation, being their number increased significantly upon rapamycin or CCCP or OA/AA treatment (Fig. 3a, b) (Mean ± SEM: SPLICS$_S$- P2A$^{LY-MT}$ CTRL 95.9 ± 6.7 $n = 30$; SPLICS$_S$- P2A$^{LY-MT}$ + Rapamycin 118.7 ± 8.2 $n = 38$ *$p \leq 0.05$,; SPLICS$_S$- P2A$^{LY-MT}$ + CCCP 143.1 ± 6.7 $n = 33$ **$p \leq 0.01$; SPLICS$_S$- P2A$^{LY-MT}$ + OA/AA 130.9 ± 10.4 $n = 27$ *$p \leq 0.01$). No effect was

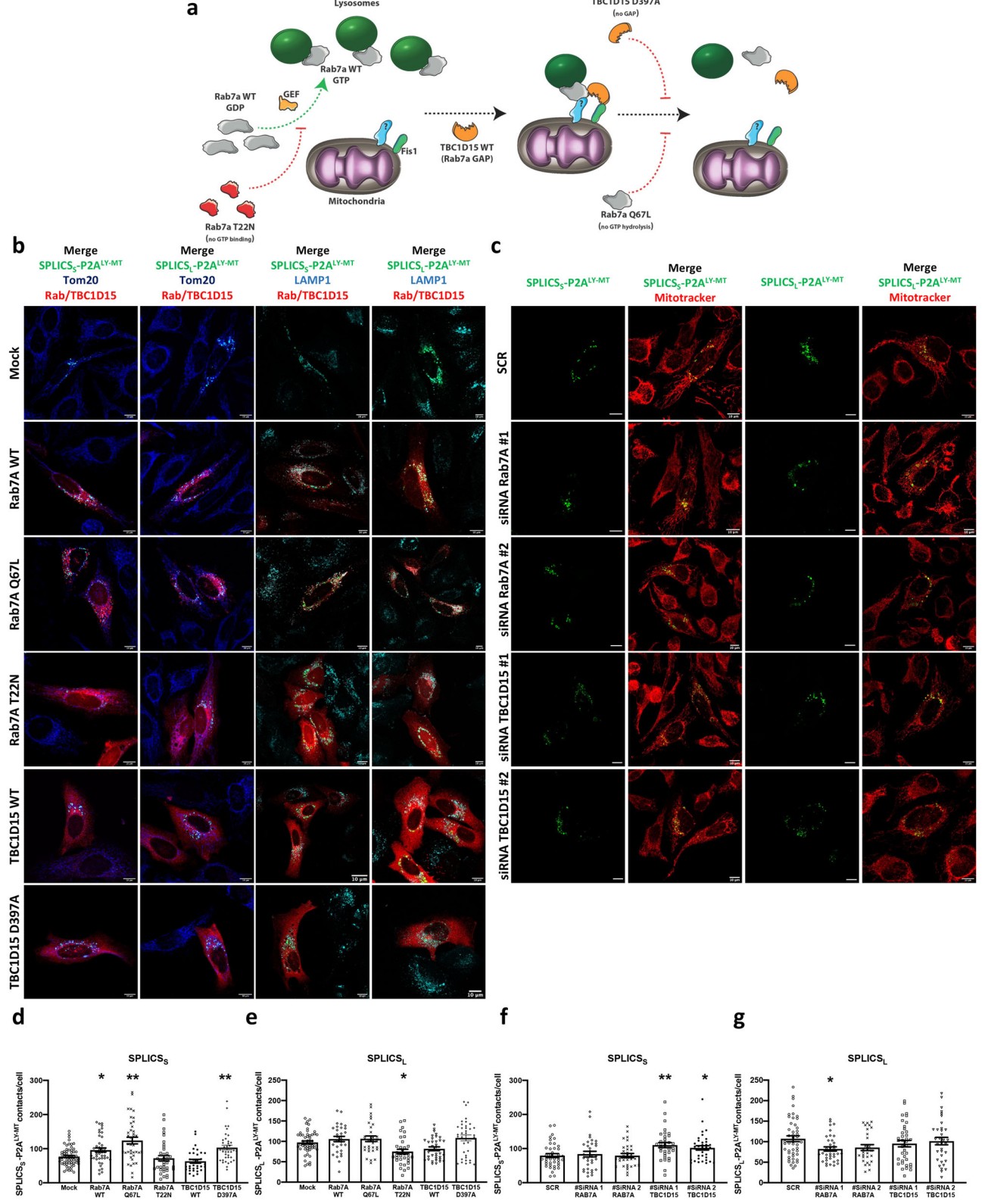

observed on long Ly-Mt interactions in both autophagy and mitophagy conditions (Fig. 3c, d) (SPLICS$_L$- P2A$^{LY-MT}$ CTRL 125.1 ± 9.6 $n$ = 35; SPLICS$_L$- P2A$^{LY-MT}$ + Rapamycin 128.3 ± 10.6 $n$ = 36; SPLICS$_L$- P2A$^{LY-MT}$ + CCCP 150.9 ± 9.6 $n$ = 36; SPLICS$_L$- P2A$^{LY-MT}$ + OA/AA 135.7 ± 10.4 $n$ = 31). Curiously, the HBSS treatment did not alter either short or long Ly-Mt interactions (Mean ± SEM: SPLICS$_S$- P2A$^{LY-MT}$ + HBSS 90.2 ± 9.1 $n$ = 30; SPLICS$_L$- P2A$^{LY-MT}$ + HBSS 143.5 ± 11.4 $n$ = 31) (Fig. 3a–d). These data indicated that only Ly-Mt contact sites occurring at ≈4 nm are

engaged during the autophagy and mitophagy activation and that our SPLICS- P2A$^{LY-MT}$ probe can discriminate the two types of contacts under these conditions.

## Lysosome–mitochondria contact sites mediated Ca$^{2+}$ transfer between the two organelles

Mitochondria and lysosomes are both important and recognized players in intracellular Ca$^{2+}$ dynamics and their dysfunction has been

**Fig. 2 | Proof-of-concept characterization of the Ly–Mt reporter. a** Cartoon of lysosome-mitochondria interface modulation by of Rab7/TBC1D15 interaction. Rab7aT22N is a constitutively GDP-binding Rab7A mutant; TBC1D397A is constitutively inactive GTPase activating mutant and Rab7aQ67L is a constitutively GTP-binding Rab7A (Q67L). **b** Representative Z-projection merge images of HeLa cells transfected with SPLICS$_{S/L}$- P2A$^{LY-MT}$ alone or plus Rab7A WT or Rab7A Q67L or Rab7A T22N or TBC1D15 WT or TBC1D15 D397A. In merge panels, SPLICS$_{S/L}$- P2A$^{LY-MT}$ (488 nm), mCherry -Rab7A WT or -Rab7A Q67L or -Rab7A T22N (568 nm) and c-myc -TBC1D15 WT or -TBC1D15 D397A (594 nm) (red signal) are shown, mitochondria or lysosomes were detected by anti-Tom20 or anti-LAMP1 (405 nm) (blue and cyan fluorescence in pseudocolor, respectively). **c** Representative Z-projection images of HeLa cells transfected with SPLICS$_{S/L}$- P2A$^{LY-MT}$ plus scramble or siRNA Rab7A #1 or siRNA Rab7A #2 or siRNA TBC1D15 #1 or siRNA TBC1D15 #2. Mitochondria morphology was detected by MitoTracker ® Red CMXRos upon excitation at 568 nm. **d, e** Quantification of short (Mean ± SEM: SPLICS$_S$- P2A$^{LY-MT}$ + Mock 76.8 ± 3.5 $n = 60$, SPLICS$_S$- P2A$^{LY-MT}$ + Rab7A 95.9 ± 9.6 $n = 43$, SPLICS$_S$- P2A$^{LY-MT}$ + Rab7A (Q67L) 123.8 ± 9.8 $n = 39$, SPLICS$_S$- P2A$^{LY-MT}$ + Rab7A (T22N) 71.9 ± 7.7 $n = 37$, SPLICS$_S$- P2A$^{LY-MT}$ + TBC1D15 63.2 ± 5.5 $n = 33$, SPLICS$_S$- P2A$^{LY-MT}$ + TBC1D15 (D397A) 103.2 ± 7.1

$n = 38$) and long (Mean ± SEM: SPLICS$_L$- P2A$^{LY-MT}$ + Mock 97.4 ± 3.9 $n = 52$; SPLICS$_L$-P2A$^{LY-MT}$ + Rab7A WT 105.8 ± 6.5 $n = 30$, SPLICS$_L$-P2A$^{LY-MT}$ + Rab7A (Q67L) 106.4 ± 6.8 $n = 31$, SPLICS$_L$-P2A$^{LY-MT}$ + Rab7A (T22N) 75 ± 5.8 $n = 34$, SPLICS$_L$-P2A$^{LY-MT}$ + TBC1D15 WT 81.9 ± 4.6 $n = 35$, SPLICS$_L$-P2A$^{LY-MT}$ + TBC1D15 (D397A) 108.5 ± 7.2 $n = 37$) LY-MT contacts in HeLa cells overexpressing WT and mutants Rab7 and TBC1D15. **f, g** Quantification of short (mean ± SEM: SPLICS$_S$- P2A$^{LY-MT}$ + SCR 79.5 ± 5.8 $n = 40$; SPLICS$_S$- P2A$^{LY-MT}$ + siRNA Rab7A #1 83.8 ± 8.1 $n = 33$; SPLICS$_S$- P2A$^{LY-MT}$ + siRNA Rab7A #2 79.4 ± 6.2 $n = 32$; SPLICS$_S$- P2A$^{LY-MT}$ + siRNA TBC1D15 #1 110.2 ± 7.1 $n = 34$; SPLICS$_S$- P2A$^{LY-MT}$ + siRNA TBC1D15 #2 97.9 ± 6.3 $n = 40$) and long (mean ± SEM: SPLICS$_L$- P2A$^{LY-MT}$ + SCR 107.1 ± 5.8 $n = 47$; SPLICS$_L$- P2A$^{LY-MT}$ + siRNA Rab7A #1 80 ± 5.2 $n = 39$ *$p \leq 0.05$; SPLICS$_L$- P2A$^{LY-MT}$ + siRNA Rab7A #2 91 ± 7.7 $n = 30$, SPLICS$_L$- P2A$^{LY-MT}$ + siRNA TBC1D15 #1 95.3 ± 8 $n = 37$; SPLICS$_L$- P2A$^{LY-MT}$ + siRNA TBC1D15 #2 101.4 ± 9.1 $n = 34$) LY-MT contacts in HeLa cells upon downregulation of Rab7 and TBC1D15. The SPLICS dots were quantified from the 3D rendering of a complete Z-stack. Scale bar 10 μm. n= cells over 3 independent transfections. (*$p \leq 0.05$, **$p \leq 0.01$ one-way ANOVA). Source data are provided as a Source Data file.

reported in many disease conditions in which dysregulation of Ca$^{2+}$ homeostasis is also present. Pioneering studies by the Kranic group have established that Ly-Mt contacts facilitated the direct transfer of Ca$^{2+}$ from lysosome to mitochondria through the Mucolipin-1 (TRPML1) lysosomal channel[29]. Here, we took advantage of this finding to verify whether the expression of our SPLICS$_{S/L}$- P2A$^{LY-MT}$ probe could impact on the formation of Ly-Mt contact sites. First we checked whether the SPLICS$_{S/L}$- P2A$^{LY-MT}$ expression could impact on mitochondrial (Fig. 4a, b) and cytosolic (Fig. 4c, d) Ca$^{2+}$ transients induced by cell stimulation with the inositol 1,4,5 trisphosphate (InsP$_3$)-linked agonist histamine by co-expressing the recombinant Ca$^{2+}$ sensitive photoprotein aequorin targeted to the mitochondrial matrix (low affinity variant) and the cytoplasm (wt), respectively. We found no differences (Mean ± SEM: [Ca$^{2+}$]$_{mt}$, μM Mock 67.2 ± 2.8 n = 46; SPLICS$_S$-P2A$^{LY-MT}$ 62.7 ± 2.6 n = 48; SPLICS$_L$-P2A$^{LY-MT}$ 72.9 ± 2.7 n = 48; [Ca$^{2+}$]$_{Cyt}$, μM Mock Mean ± SEM: Mock 3.2 ± 0.04 $n = 47$; SPLICS$_S$-P2A$^{LY-MT}$ 3.1 ± 0.04 n = 47; SPLICS$_L$-P2A$^{LY-MT}$ 3.3 ± 0.04 n = 51), suggesting that no major perturbations in cytosolic Ca$^{2+}$ handling and in mitochondria ability to sense Ca$^{2+}$ released form the intracellular stores has occurred upon overexpression of the SPLICS$_{S/L}$- P2A$^{LY-MT}$ reporters. Then, we focused our attention of the Ly-Mt Ca$^{2+}$ transfer and to this purpose Hela cells were transfected with the mitochondrial targeted wt aequorin which covers a dynamic range of [Ca$^{2+}$], from $10^{-7}$ to $10^{-5}$ M[71]. The cells (mock and co-expressing SPLICS$_{S/L}$- P2A$^{LY-MT}$) were stimulated with ML-SA1, a selective TRPML1 activator[72], to induce lysosomal Ca$^{2+}$ release, as previously described. Figure 4e shows the traces of Ca$^{2+}$ transients generated in the mitochondrial matrix upon ML-SA1 stimulation in mock cells and in cells overexpressing SPLICS$_{S/L}$- P2A$^{LY-MT}$. We have found that the presence of SPLICS$_{S/L}$-P2A$^{LY-MT}$ did not impact on mitochondrial uptake of Ca$^{2+}$ released by lysosomes (Mean ± SEM, μM: Mock 1.306 ± 0.073 $n = 23$; SPLICS$_S$-P2A$^{LY-MT}$ 1.283 ± 0.055 n = 25; SPLICS$_L$-P2A$^{LY-MT}$ 1.154 ± 0.079 n = 26; Fig. 4f). To further validate this conclusion, we decided to measure Ca$^{2+}$ transfer under conditions in which an increase of lysosome-mitochondria interaction has been induced by the treatment with the U18666A, a Niemann-Pick type C protein 1 (NPC1) inhibitor, as previously reported[30]. Filipin III staining shown in Fig.4g revealed that U18666A treatment induced a strong cholesterol accumulation at lysosomes due to NPC1 inhibition, as expected and previously reported[30]. This cholesterol accumulation was reflected in an increased number of Ly-Mt short contacts (Mean ± SEM: SPLICS$_S$-P2A$^{LY-MT}$ CTRL 88.4 ± 8.2 $n = 21$; SPLICS$_S$-P2A$^{LY-MT}$ U18666A 139.8 ± 9.6 $n = 30$ ***$p \leq 0.001$) without affecting long ones (Mean ± SEM: SPLICS$_L$-P2A$^{LY-MT}$ CTRL 102.9 ± 10.6 $n = 24$; SPLICS$_L$-P2A$^{LY-MT}$ U18666A 127.5 ± 11.4 $n = 30$) as shown in Fig. 4h–j. No changes in cytosolic Ca$^{2+}$ transients were reported upon treatment with U18666A (Mean ± SEM, μM: CTRL 0.561 ± 0.037 n = 23; U18666A

0.589 ± 0.025 $n = 16$) (Fig. 4k, l); intriguingly, under the same conditions, we detected a significant increase in mitochondrial Ca$^{2+}$ transients upon stimulation with ML-SA1 (Mean ± SEM, μM: CTRL 0.968 ± 0.047 $n = 32$; U18666A 1.240 ± 0.042 $n = 32$ ****$p \leq 0.0001$) (Fig. 4m, n). These results support the existence of localized Ca$^{2+}$ microdomains generated upon ML-SA1 induced Ca$^{2+}$ release from lysosomes that are sensed by mitochondria and depend on lysosome-mitochondria tethering. This observation is very challenging since it is well established that lysosomes and mitochondria are in contact[7,73], but little is known on the functionality of these contacts. Considering this and the finding that Rab7A and TBC1D15 manipulation modulated the number of Ly-Mt contact sites (see above and Fig. 2), we explored their possible effect on lysosomal-mitochondrial Ca$^{2+}$ transfer. Notably, the overexpression of Rab7A (Q67L) and TBC1D15 (D397A) mutants induced a strong increase in mitochondrial Ca$^{2+}$ transients, while the wt forms and Rab7A (T22N) did not induce any changes (Mean ± SEM, μM: Mock 0.995 ± 0.043 $n = 58$; Rab7A WT 1.060 ± 0.045 $n = 54$; Rab7A (Q67L) 1.211 ± 0.042 $n = 60$ **$p \leq 0.01$; Rab7A (T22N) 1.125 ± 0.039 $n = 60$; TBC1D15 WT 1.012 ± 0.041 $n = 60$; TBC1D15 (D397A) 1.234 ± 0.057 $n = 54$ **$p \leq 0.01$) (Fig. 4o, p). Similarly, the silencing of TBC1D15 with two different siRNAs caused a significative Ca$^{2+}$ transient rise, while the silencing of Rab7A protein did not alter them (Mean ± SEM: SCR 0.955 ± 0.044 $n = 38$; siRNA Rab7A #1 1.095 ± 0.057 $n = 36$; siRNA Rab7A #2 1.072 ± 0.045 $n = 38$; siRNA TBC1D15 #1 1.192 ± 0.059 $n = 42$ **$p \leq 0.01$; siRNA TBC1D15 #2 1.193 ± 0.075 $n = 39$ *$p \leq 0.05$) (Fig. 4q, r). Taken together, these data indicate that Ca$^{2+}$ transfer from lysosomes to mitochondria is strictly dependent on their tethering and on the contact sites occurring in the short 4 nm range.

## Detection of lysosome-mitochondria contact sites in vivo

The experiments shown above, clearly demonstrated the ability of the SPLICS$_{S/L}$-P2A$^{LY-MT}$ reporter to detect lysosome-mitochondria contacts and their changes in cultured cell lines. We therefore decided to tested it on more polarized cells as well as in vivo. To this aim we have chosen three different approaches: (i) mouse primary cortical neurons, (ii) Rohon−Beard (RB) sensory neurons of living Zebrafish animals and (iii) wing sensory neurons of living *Drosophila* flies. As shown in Fig. 5a primary cortical neurons from wt mice were isolated and transfected with the SPLICS$_S$-P2A$^{LY-MT}$ reporter and counterstained with Hoechst (405 nm) to detect nuclei and β-tubulin III (647 nm) as neuronal marker. Ly-Mt contact sites were clearly evident in the soma (Fig. 5a, red arrowheads) and in the axon (Fig. 5a, white arrows) of the transfected neuron. Interestingly, quantification of the number of Ly-Mt contacts revealed that most of them are present in the soma (Fig,5b, Mean ± SEM: SPLICS$_S$-P2A$^{LY-MT}$ 20.19 ± 2.17 $n = 34$, soma; 0.23 ± 0.035 $n = 34$,

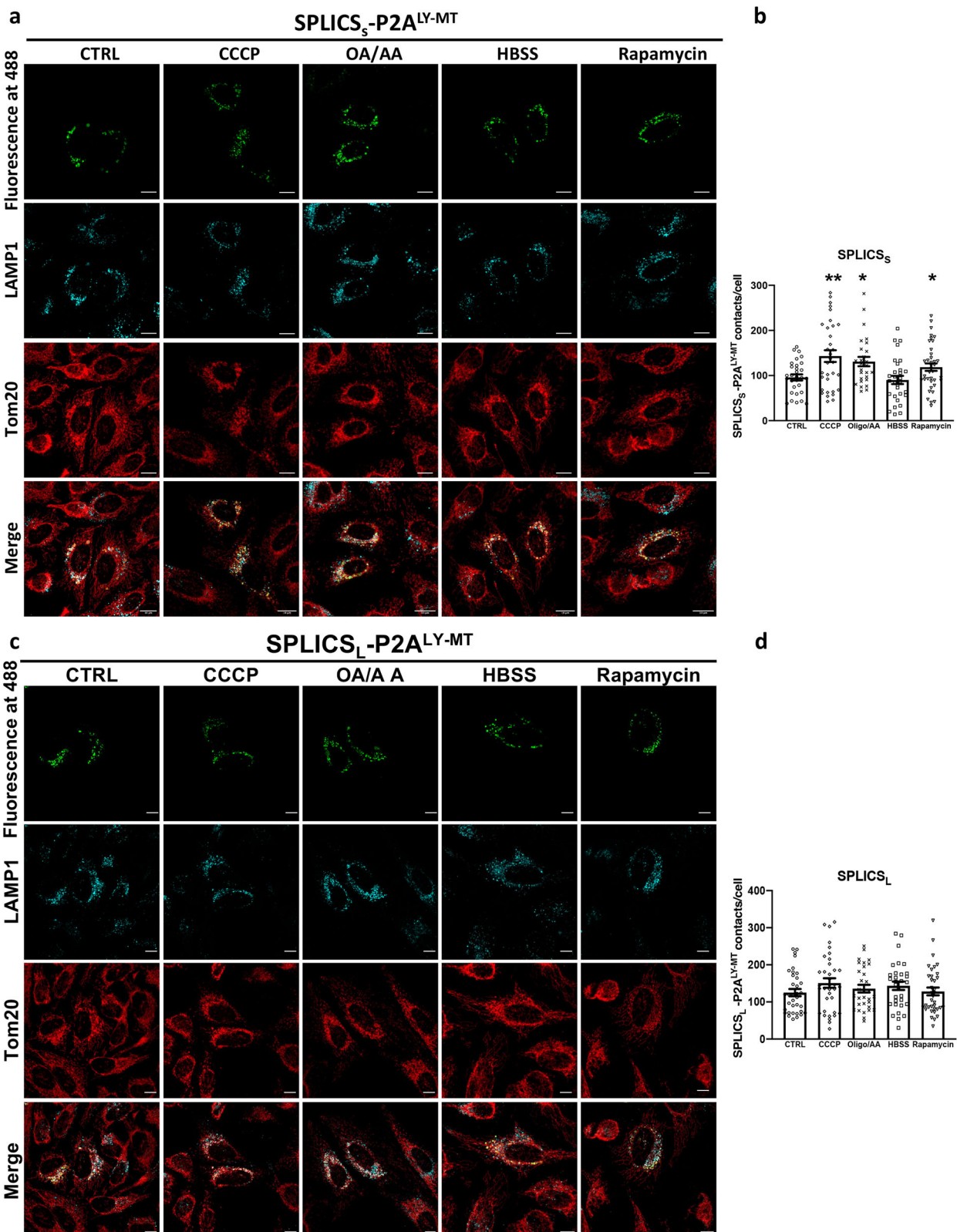

axon). Live imaging of Ly-Mt contact sites in zebrafish was performed by injecting the pT2-DsRed-UAS-SPLICS$_S$-P2A$^{LY-MT}$ construct into fertilized eggs of s1102t:GAL4 fish, selectively expressing GAL4 in the RB neurons. At 24 hpf high-resolution images of anesthetized embryos were acquired in a confocal microscope. As shown in Fig. 5c, Ly-Mt contacts appeared either in the soma (red arrowheads) or distributed in the axons of RB neurons (white arrows) marked in red thanks to the

expression of a cytosolic DsRed. We were not able to reliably quantify contacts in the soma since their high density makes the quantification potentially underestimated, axonal contacts in 50 µm of axonal tracts were instead quantified (Fig. 5d, Mean ± SEM: SPLICS$_S$-P2A$^{LY-MT}$ 24.53 ± 2.607 $n = 15$). Finally, we also generated transgenic flies expressing the SPLICS$_S$-P2A$^{LY-MT}$ under control of an UAS enhancer sequence. Figure 5e shows the pattern of probe expression under

**Fig. 3 | Effects of autophagy and mitophagy stimuli on LY-MT contacts.**
**a** Representative Z-projection images of HeLa cells transfected with SPLICS$_S$- P2A$^{LY-MT}$ in untreated condition (CTRL) or treated with CCCP or oligomycinA/ antimycinA (OA/AA) to induces mitophagy or with HBSS or rapamycin to induce starvation. SPLICS$_S$- P2A$^{LY-MT}$ (488 nm), mitochondria and lysosomes were detected by anti-Tom20 or anti-LAMP1 upon excitation at 594 and 405 nm, respectively. **b** Quantification of short LY-MT contacts in untreated and treated HeLa cells (Mean ± SEM: SPLICS$_S$- P2A$^{LY-MT}$ CTRL 95.9 ± 6.7 $n = 30$, SPLICS$_S$- P2A$^{LY-MT}$ + CCCP 143.1 ± 6.7 $n = 33$, SPLICS$_S$- P2A$^{LY-MT}$ + OA/AA 130.9 ± 10.4 $n = 27$, SPLICS$_S$- P2A$^{LY-MT}$ + HBSS 90.2 ± 9.1 $n = 30$, SPLICS$_S$- P2A$^{LY-MT}$ + Rapamycin 118.7 ± 8.2 $n = 38$). **c** Representative Z-projection images of HeLa cells transfected with SPLICS$_L$- P2A$^{LY-MT}$ in untreated condition (CTRL) or treated with CCCP or oligomycin A/ antimycin A (OA/AA) to induces mitophagy or with HBSS or rapamycin to induce starvation. SPLICS$_L$- P2A$^{LY-MT}$ (488 nm), mitochondria and lysosomes were detected by anti-Tom20 or anti-LAMP1 upon excitation at 594 and 405 nm, respectively. **d** Quantification of long (SPLICS$_L$- P2A$^{LY-MT}$ CTRL 125.1 ± 9.6 $n = 35$, SPLICS$_L$- P2A$^{LY-MT}$ + CCCP 150.9 ± 9.6 $n = 36$, SPLICS$_L$- P2A$^{LY-MT}$ + OA/AA 135.7 ± 10.4 $n = 31$, SPLICS$_L$- P2A$^{LY-MT}$ + HBSS 143.5 ± 11.4 $n = 31$, SPLICS$_L$- P2A$^{LY-MT}$ + Rapamycin 128.3 ± 10.6 $n = 36$) LY-MT contacts in untreated and treated HeLa cells. The SPLICS dots were quantified from the 3D rendering of a complete Z-stack. Scale bar 10 μm. $n$ = cells over 3 independent transfections. (*$p ≤ 0.05$, **$p ≤ 0.01$ one-way ANOVA). Source data are provided as a Source Data file.

control of the pan neuronal driver Appl-Gal4. Ly-Mt contact sites were clearly evident in the soma (red arrowheads) and in the axon (white arrows). Also in this case those in 50 μm of axon could be reliably quantified (Fig. 5f) (Mean ± SEM: SPLICS$_S$- P2A$^{LY-MT}$ 18.81 ± 0.6471 $n = 36$). Altogether these experiments demonstrate that the SPLICS$_S$-P2A$^{LY-MT}$ can efficiently detect Ly-Mt contact sites in vivo.

## α-Synuclein decreases the number of short lysosome-mitochondria contact sites
Once established that our tool was reliable and powerful in monitoring changes in lysosome-mitochondria contact sites we decided to investigate an important biological issue. Considering that defects in lysosome and mitochondria function and communication have been reported to occur in neurodegenerative processes[14] we decided to explore whether the manipulation of the expression levels of the PD-related protein α-syn could impact on Ly-Mt tethering and Ly-Mt Ca$^{2+}$ transfer. We had previously demonstrated that α-syn plays a role on ER-mitochondria tethering and thus impacts on Ca$^{2+}$ transfer and cell bioenergetics[74–77]. Here we investigated the role of α-syn WT and its two mutants, A30P and A53T, on Ly-Mt interactions by co-expressing them with the SPLICS$_S$- P2A$^{LY-MT}$. We focused on contact sites occurring in the short-range since, based on our data, they are those involved in the process of Ca$^{2+}$ transfer from lysosome to mitochondria. Intriguingly, we found that α-syn WT, A30P and A53T drastically decreased the number of short contact sites compared to Mock cells (Mean ± SEM: SPLICS$_S$- P2A$^{LY-MT}$ + Mock 87.84 ± 4.46 $n = 64$; SPLICS$_S$- P2A$^{LY-MT}$ + WT 50.67 ± 2.77 $n = 50$; SPLICS$_S$- P2A$^{LY-MT}$ + A30P 54.60 ± 3.39 $n = 70$; SPLICS$_S$- P2A$^{LY-MT}$ + A53T 56.35 ± 2.69 $n = 69$; ****$p ≤ 0.0001$) (Fig. 6a, b). To exclude that α-syn overexpression could modify the total lysosomal content, and eventually impact on Ly-Mt short contact sites for this reason, we quantified lysosomes abundance by immunocytochemistry and Western blotting analysis upon their labeling with an anti-LAMP1 antibody (Fig. 6c–f). The results of this analysis revealed no differences in the lysosomal pool under the different conditions (Mean ± SEM: Mock 92.04 ± 7.27 $n = 37$; WT 94.41 ± 4.66 $n = 27$; A30P 89.92 ± 6.63 $n = 26$; A53T 115.7 ± 11.29 $n = 16$). Then, to evaluate whether the decrease in Ly-Mt interactions observed in α-syn overexpressing cells could impact on mitochondrial Ca$^{2+}$ transients upon TRPML1-mediated lysosomal Ca$^{2+}$ release, we co-transfected HeLa cells with wt mitochondrial aequorin[78] and monitored Ca$^{2+}$ concentration in the mitochondrial matrix upon ML-SA1 stimulation. The results suggest that the overexpression of α-syn WT (documented in Supplementary Fig. 4) slightly but not significantly decreased the mitochondrial Ca$^{2+}$ uptake in these conditions (Mean ± SEM, μM: Mock 1.230 ± 0.069 $n = 37$; α-syn WT 1.130 ± 0.059 $n = 39$) (Fig. 6g, h). Since ML-SA1 has been reported to induce Ca$^{2+}$ entry from the extracellular ambient via SOCE activation[79] and α-syn as well has been reported to impact on Ca$^{2+}$ entry[80] to better appreciate the contribute of Ca$^{2+}$ release from lysosomes we performed the same experiment in the absence of extracellular Ca$^{2+}$ (Fig. 6i). Notably, the mitochondrial Ca$^{2+}$ transients generated by ML-SA1 in KRB medium containing 100 μM EGTA were reduced in respect to those generated in the presence of extracellular Ca$^{2+}$, i.e., they are in the range of 0.5 μM. Under these

conditions it was possible to detect that in cells overexpressing α-syn WT mitochondrial Ca$^{2+}$ levels were significantly reduced compared to control cells (Mean ± SEM, μM: Mock 0.46 ± 0.03 $n = 10$; α-syn WT 0.34 ± 0.03 $n = 10$ **$p ≤ 0.01$) (Fig. 6i, j). To exclude the possibility that the observed differences were due to differences in Ca$^{2+}$ release from InsP3-sensitive intracellular compartments, after ML-SA1 stimulation, the cells were stimulated with histamine. As expected, upon histamine addition mitochondrial Ca$^{2+}$ uptake was observed, but no differences were detected between control and α-syn WT overexpressing cells (Mean ± SEM, μM: Mock 5.13 ± 0.83 $n = 9$; α-syn WT 5.55 ± 0.28 $n = 9$) (Fig. 6k). Altogether these results suggested that α-syn WT affects lysosome-mitochondria interaction by reducing the number of their contact sites and reduces mitochondrial Ca$^{2+}$ uptake upon lysosomal Ca$^{2+}$ release, thus possibly interfering with the dissipation of cytosolic microdomains of local Ca$^{2+}$ concentration generated by the opening of TRPLM1 channel.

## α-Synuclein overexpression counteracts the increase of Ly-Mt short interactions induced by lysosomal cholesterol accumulation
Considering that both dysregulation in cholesterol flux and α-syn clearance play key roles in PD pathogenesis[81,82] and that, based on our findings, α-syn reduces, but cholesterol accumulation increases, the interplay between lysosomes and mitochondria, we analyzed the Ly-Mt interactions by combining the two conditions. HeLa cells were co-transfected with α-syn WT or A30P/A53T mutants and SPLICS$_S$-P2A$^{LY-MT}$ and then, incubated with U18666A. According to the results reported above (see Fig. 4h, i), U18666A treatment drastically increased the short contacts in control cells (Mean ± SEM: Untreated SPLICS$_S$- P2A$^{LY-MT}$ + Mock 60.57 ± 4.73 $n = 43$; U18666A treated SPLICS$_S$- P2A$^{LY-MT}$ + Mock 83.49 ± 4.09 $n = 39$ ****$p ≤ 0.0001$, Fig. 7a, b), however, its action was completely blunted in presence of overexpressed α-syn WT, A30P and A53T (Mean ± SEM: Untreated: SPLICS$_S$- P2A$^{LY-MT}$ + WT 35.15 ± 3.83 $n = 31$; SPLICS$_S$- P2A$^{LY-MT}$ + A30P 38.07 ± 3.78 $n = 36$; SPLICS$_S$- P2A$^{LY-MT}$ + A53T 31.83 ± 3.39 $n = 38$. U18666A treated: SPLICS$_S$- P2A$^{LY-MT}$ + WT 41.06 ± 4.09 $n = 36$; SPLICS$_S$- P2A$^{LY-MT}$ + A30P 43.58 ± 3.30 $n = 31$; SPLICS$_S$- P2A$^{LY-MT}$ + A53T 38.08 ± 3.37 $n = 38$) (Fig. 7a, b). Filipin III staining revealed that U18666A treatment induced a strong cholesterol accumulation at lysosomes due to NPC1 inhibition regardless of the presence of α-syn WT (Supplementary Fig. 5). Notably, no relevant differences were reported comparing the wild-type form of α-syn with the two mutants. To elucidate functional consequences, we decided to evaluate the mitochondrial Ca$^{2+}$ transients upon TRPLM1 activation in control cells and in cells overexpressing α-syn WT, both incubated with U18666A. Consistently with the effect observed on Ly-Mt contact sites, the overexpression of α-syn WT decreases mitochondrial Ca$^{2+}$ transients even in the cells incubated with U18666A (Mean ± SEM, μM: Mock 1.954 ± 0.116 $n = 25$; α-syn WT 1.567 ± 0.082 $n = 39$ **$p ≤ 0.01$) (Fig. 7c, d). These data demonstrated that α-syn WT abolished the upregulation of Ly-Mt interactions induced by lysosomal cholesterol accumulation and this was also reflected in a reduced mitochondrial Ca$^{2+}$ uptake. Notably, α-syn WT impinged on Ly-Mt tethering also under HBSS induced starvation, which per se does not impact on Ly-Mt

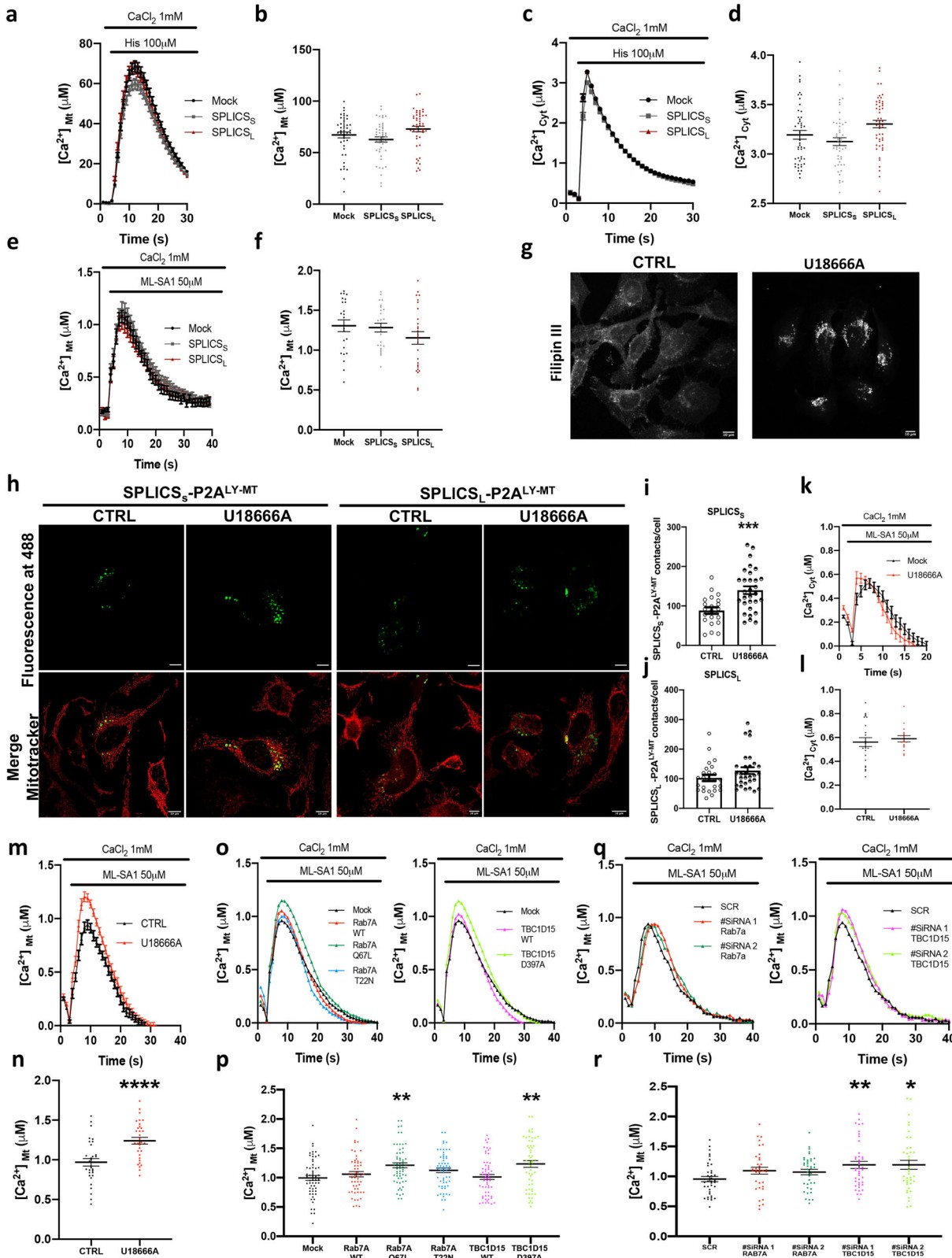

contact sites and α-syn overexpressing cells, upon HBSS treatment displayed an impaired autophagic capacity that was not observed under basal condition (Supplementary Fig. 6).

## α-Synuclein is involved in TFEB cellular localization

In many forms of PD, accumulation of α-syn, which in turn leads to its improper aggregation, is a common hallmark for the disease progression[83]. In this respect, lysosomal activity carries out a crucial role in controlling the clearance of α-syn meaning that lysosomal biogenesis could be stimulated to sustain this demand[52]. The Transcription Factor EB (TFEB) is a master gene for lysosomal biogenesis sensitive to cellular homeostasis alterations such as amino acids deprivation and TFEB-mediated autophagy has been reported to rescue midbrain dopamine neurons from α-syn toxicity[52]. Under

**Fig. 4 | Lysosomes–mitochondria contact sites mediated calcium transfer.**
**a** Mitochondrial and (**c**) Cytosolic $Ca^{2+}$ transients generated upon histamine (his 100 µM) stimulation in HeLa cells expressing the $Ca^{2+}$ sensitive mtAEQmut or cytAeqwt probes alone (Mock) or co-expressing $SPLICS_{S/L}$- $P2A^{LY-MT}$ and mtAEQmut or cytAeqwt probes, respectively; **b, d** average peak values, mean ± SEM: $[Ca^{2+}]_{mt}$, µM Mock 67.2 ± 2.8 $n = 46$; $SPLICS_S$-$P2A^{LY-MT}$ 62.7 ± 2.6 $n = 48$; $SPLICS_L$-$P2A^{LY-MT}$ 72.9 ± 2.7 $n = 48$; $[Ca^{2+}]_{Cyt}$, µM Mock Mean ± SEM: Mock 3.2 ± 0.04 $n = 47$; $SPLICS_S$-$P2A^{LY-MT}$ 3.1 ± 0.04 $n = 47$; $SPLICS_L$-$P2A^{LY-MT}$ 3.3 ± 0.04 $n = 51$. **e** Mitochondrial $Ca^{2+}$ transients generated upon application of ML-SA1, a selective activator of lysosomal TRPML1, in HeLa cells co-expressing $SPLICS_{S/L}$- $P2A^{LY-MT}$ and mtAeqwt probes; Mock refers to Hela cells expressing mtAeqwt alone, **f** Average peak values, mean ± SEM: $[Ca^{2+}]_{mt}$ µM, Mock 1.306 ± 0.073 $n = 23$; $SPLICS_S$-$P2A^{LY-MT}$ 1.283 ± 0.055 $n = 25$; $SPLICS_L$-$P2A^{LY-MT}$ 1.154 ± 0.079 $n = 26$. **g** Representative images of HeLa cells untreated (CTRL) or treated (U18666A) stained with Filipin III to detect intracellular cholesterol accumulation. **h** Representative Z-projection images of HeLa cells transfected with $SPLICSS_{S/L}$ $P2A^{LY-MT}$ in untreated condition (CTRL) or treated with U18666A. $SPLICS_{S/L}$- $P2A^{LY-MT}$ were represented by fluorescent "dots" upon excitation at 488 nm, mitochondria morphology was detected by MitoTracker ® Red CMXRos upon excitation at 568 nm. **i, j** Quantification of short (Mean ± SEM: $SPLICS_S$-$P2A^{LY-MT}$ CTRL 88.4 ± 8.2 $n = 21$; $SPLICS_S$-$P2A^{LY-MT}$ U18666A 139.8 ± 9.6 $n = 30$) and long (Mean ± SEM: $SPLICS_L$-$P2A^{LY-MT}$ CTRL 102.9 ± 10.6 $n = 24$; $SPLICS_L$-$P2A^{LY-MT}$ U18666A 127.5 ± 11.4 $n = 30$) LY-MT contacts in untreated and treated HeLa

cells. The SPLICS dots were quantified from the 3D rendering of a complete Z-stack, mean ± SEM. **k, l** Cytosolic $Ca^{2+}$ transients in untreated (CTRL) or treated with U18666A HeLa cells upon ML-SA1 stimulation and average peak values, mean ± SEM: $[Ca^{2+}]_{Cyt}$, µM CTRL 0.561 ± 0.037 $n = 23$; U18666A 0.589 ± 0.025 $n = 16$. **m, n** Mitochondrial $Ca^{2+}$ transients in untreated (CTRL) or treated with U18666A HeLa cells upon ML-SA1 stimulation and average peak values, mean ± SEM: $[Ca^{2+}]_{mt}$, µM CTRL 0.968 ± 0.047 $n = 32$; U18666A 1.240 ± 0.042 $n = 32$. **o, p** Mitochondrial $Ca^{2+}$ transients generated upon ML-SA1 stimulation in HeLa cells transfected with mtAeqwt and empty pcDNA3 (Mock) or Rab7A WT or Rab7A Q67L or Rab7A T22N or TBC1D15 WT or TBC1D15 D397A and average peak values, mean ± SEM: $[Ca^{2+}]_{mt}$, µM Mock 0.995 ± 0.043 $n = 58$; Rab7A WT 1.060 ± 0.045 $n = 54$; Rab7A (Q67L) 1.211 ± 0.042 $n = 60$; Rab7A (T22N) 1.125 ± 0.039 $n = 60$; TBC1D15 WT 1.012 ± 0.041 $n = 60$; TBC1D15 (D397A) 1.234 ± 0.057 $n = 54$. **q, r** Mitochondrial $Ca^{2+}$ transients generated upon ML-SA1 stimulation in HeLa cells transfected with mtAeqwt and scramble or siRNA Rab7A #1 or siRNA Rab7A #2 or siRNA TBC1D15 #1 or siRNA TBC1D15 #2 and average peak values, mean ± SEM: $[Ca^{2+}]_{mt}$, µM SCR 0.955 ± 0.044 $n = 38$; siRNA Rab7A #1 1.095 ± 0.057 $n = 36$; siRNA Rab7A #2 1.072 ± 0.045 $n = 38$; siRNA TBC1D15 #1 1.192 ± 0.059 $n = 42$; siRNA TBC1D15 #2 1.193 ± 0.075 $n = 39$. Scale bar 10 µm. $n$ = cells (**l, j**) or coverslips (**b, d, f, l, n, p, r**) examined over 3 independent experiments. In (**i, n**) ***$p \le 0.001$, ****$p \le 0.0001$ unpaired two-tailed $t$ test; in (**p, r**) *$p \le 0.05$, **$p \le 0.01$ one-way ANOVA. Source data are provided as a Source Data file.

physiological conditions the mammalian target of rapamycin complex 1 (mTORC1) phosphorylates the cytosolic TFEB on the lysosomal membrane, inhibiting its transcriptional activity[59,62,84,85]. mTORC1 inactivation reduces the phosphorylated TFEB pool, and meanwhile, the activation of the $Ca^{2+}$ sensitive phosphatase calcineurin contributes to dephosphorylate TFEB inducing its nuclear translocation and activation[63,64]. Calcineurin activation is directly dependent on lysosomal $Ca^{2+}$ release, which can locally mediate TFEB dephosphorylation and activate it[63]. Furthermore, it has been recently shown that lysosomal $Ca^{2+}$ release also promotes specific inactivation of RagC/D GTPases[86,87], thus leading to TFEB nuclear translocation via impairment of mTORC1-mediated phosphorylation[65,88]. Considering that in the presence of α-syn the amount of $Ca^{2+}$ taken up by mitochondria upon release from lysosomes is reduced, implying that cytosolic $Ca^{2+}$ levels in the proximity of the site of release could be increased, we decided to evaluate whether TFEB nuclear translocation was increased in α-syn WT overexpressing cells. To explore this possibility, we transfected HeLa cells with the expression vector for TFEB-GFP alone or together with that for the expression of α-syn WT (Fig. 8a, b). Under basal conditions (Fig. 8a), TFEB has a prevalent cytosolic localization. However, co-expression of α-Syn WT promoted a significant increase in TFEB nuclear translocation (Fig. 8b). To further corroborate these data, we also transfected α-Syn WT in a widely used HeLa cells clone stably expressing chimeric TFEB-GFP[65]. As shown in (Fig. 8c) and quantified in (Fig. 8d), TFEB nuclear localization was significantly enhanced in cells co-expressing α-Syn.

Notably, the massive nuclear translocation of the lysosome-associated pool of TFEB induced by α-Syn WT could be fully reverted by treatment with the intracellular $Ca^{2+}$ chelator BAPTA (Fig. 8e, f), highlighting the key role played by calcium in the microdomain between mitochondria and lysosomes. The results shown above are in agreement with a previous report showing a bi-temporal response of α-Syn overexpression on TFEB nuclear translocation, where an early presumably protective response to α-syn overexpression associated with increased TFEB translocation into the nucleus was first observed, and, at a later stage, reduced expression of TFEB in the nuclear fraction (sequestered into the cytoplasm by α-Syn) led to impaired autophagy[52]. Our data demonstrate that when α-Syn is expressed at an amount that does not impige on autophagy (see Supplementary Fig. 5), the observed activation of TFEB is the consequence of α-Syn action on lysosome-mitochondria contact sites that leads to reduced $Ca^{2+}$

buffering by mitochondria, possibly increasing the cytosolic $Ca^{2+}$ microdomain close to the lysosomes surface and leading to $Ca^{2+}$-dependent calcineurin activation or/and TRPML1-dependent inhibition of RagC/D GTPases. Potentially both of these pathways could well explain the mechanism by which nuclear TFEB translocation early protective phase is triggered.

## Discussion

We have generated a genetically encoded reporter for the analysis of the lysosome-mitochondria contact sites ($SPLICS_{S/L}$-$P2A^{MT-LY}$) opening to the possibility to assess them in vitro and in vivo, in an easy, single-step and semi-automated manner[35–38]. Importantly, $SPLICS_{S/L}$-$P2A^{MT-LY}$ has unveiled the existence of functionally distinct short- and long-range lysosome-mitochondria interactions which differently responded to the activation of autophagy and mitophagy pathways and promptly and selectively reacted to changes in the levels of specific known tethering/untethering factors. Overexpression of the WT Rab7A, of the constitutively active Rab7 (Q67L)-GTP mutant or of the constitutively inactive TBC1D15 (D397A) GAP-domain mutant significantly increased short- range lysosome-mitochondria interactions (confirming previous findings[8,20,29] and further supporting the reliability of the SPLICS reporters), leaving the long-range interactions essentially unaffected. The same phenotype could be observed upon downregulation of the endogenous untether TBC1D15, while decreased levels of endogenous Rab7 showed no significant effect on either short- and long-range contact sites, suggesting that the remaining pool of Rab7 protein could be sufficient to support the tethering. Short-range lysosome-mitochondria contact sites were increased upon mitophagy induction with CCCP or oligomycin/antimycin treatment while rapamycin, but not HBSS, treatment only slightly increased them suggesting the intriguing possibility that different starvation conditions could well result in different engagement of this contact interface, which is also different from the one engaged under basal conditions. Long-range lysosome-mitochondria contact sites remained unaltered under the same conditions, suggesting that these two types of contact sites may respond differently to the same stimuli, implying their involvement in different functional pathways. Functional assessment of local lysosome-to-mitochondria $Ca^{2+}$ transfer revealed no artificial tethering caused by the SPLICS reporters, rather confirmed the action of the above-mentioned tethers/untethers, which exquisitely mirrored the foreseen action on the lysosome-

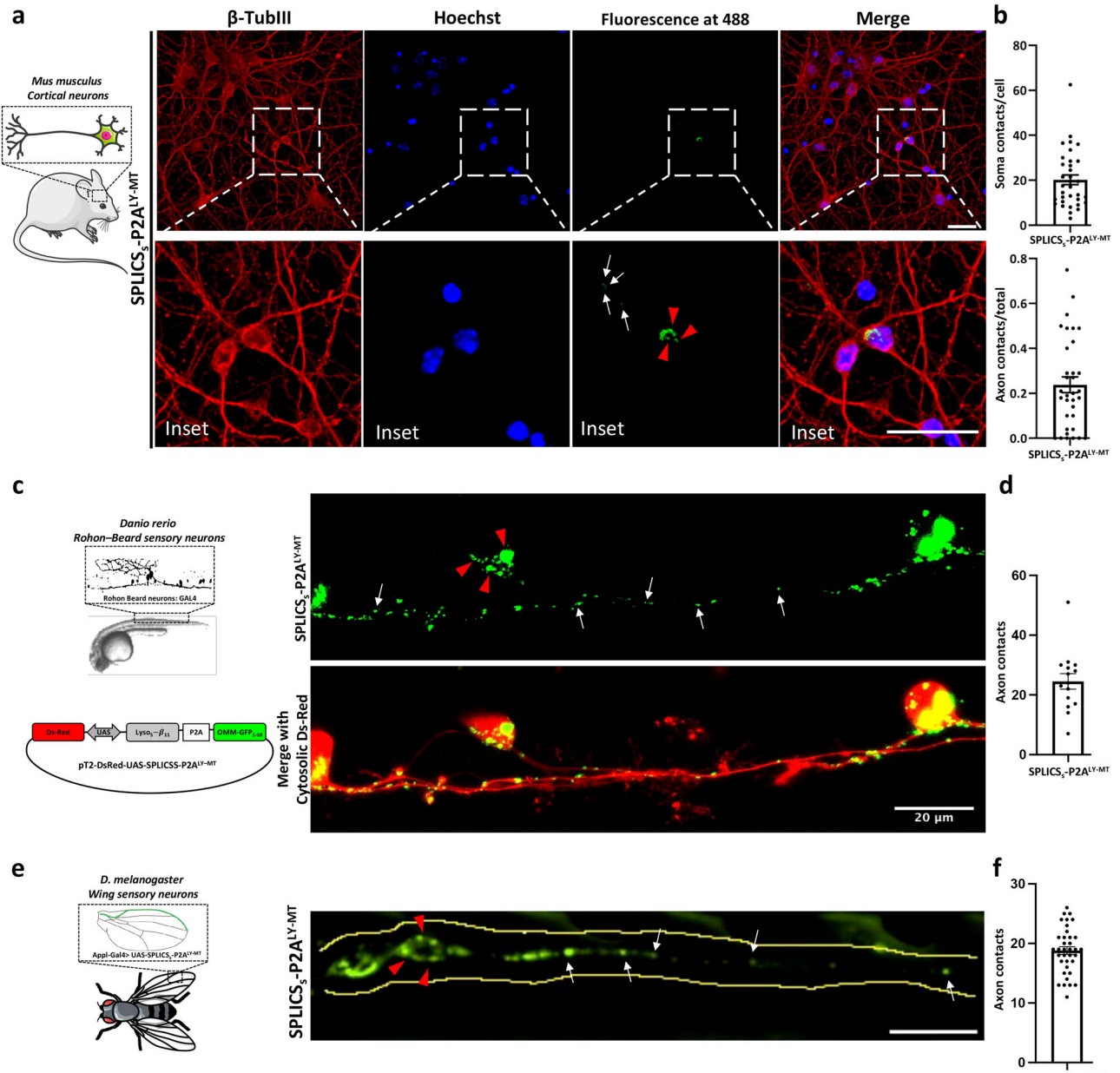

**Fig. 5 | Detection of lysosome-mitochondria contact sites in vivo.**
**a** Representative images (Z-projections from a complete Z-stack) of Mouse Primary Cortical Neurons (MPCNs) isolated from wt mice, stained with an anti β-TubIII (647 nm) and Hoechst (405 nm), with SPLICS$_S$- P2A$^{LY-MT}$ probe (488 nm). Scale bar 25 μm. **b** Quantification of the Ly-Mt contacts was performed from the 3D rendering of a complete Z-stack. Data is shown as mean ± SEM dots/cell (left) or dots in axons/total dots (SPLICS$_S$-P2A$^{LY-MT}$ 20.19 ± 2.17 $n$ = 34, soma; 0.23 ± 0.035 $n$ = 34, axon, $n$ = cells examined over 4 independent experiments). **c** Live imaging of Ly-mt contact sites in zebrafish RB neurons, representative confocal images of SPLICS$_S$-P2A$^{LY-MT}$ in RB neurons of 24 hpf s1102t:GAL4 living embryos injected with the pT2-DsRed-UAS-SPLICS$_S$-P2A$^{LY-MT}$ construct. Rostral is on the right, dorsal on the top.
**d** Quantification of the Ly-Mt contacts was performed from the 3D rendering of a complete Z-stack, mean ± SEM: SPLICS$_S$-P2A$^{LY-MT}$ 24.53 ± 2.607 $n$ = 15 cells over at least 3 independent injections. **e** Live imaging and (**f**) quantification of Ly-mt contact sites in *D. melanogaster* wing sensory neurons ($n$ = 36 wings) from 2-day old flies (Mean ± SEM: SPLICS$_S$- P2A$^{LY-MT}$ 18.81 ± 0.6471 $n$ = 36). Dashed line: boundaries of the neuronal tract. The soma and the axon are indicated with red arrowheads and white arrows, respectively. Data points represent individual SPLICS puncta from 50 μm of axons. "Parts of the figure were drawn by using pictures from Servier Medical Art. Servier Medical Art by Servier is licensed under a Creative Commons Attribution 3.0 Unported License (https://creativecommons.org/licenses/by/3.0/)." Source data are provided as a Source Data file.

mitochondria contact sites. Accordingly, global and local Ca$^{2+}$ handling was not affected by SPLICS reporters expression (Fig. 4a–f) per se, while the promotion of the lysosome-mitochondria interface by U18666A[30], or by the overexpression of the constitutively active Rab7 (Q67L)-GTP mutant or the TBC1D15 (D397A) GAP-domain mutant, enhanced the amount of Ca$^{2+}$ sensed by mitochondria upon its release from lysosomes (Fig. 4m–p). According to the data above, down-regulation of Rab7 didn't show any effect on lysosome-to-

mitochondria Ca$^{2+}$-transfer while knockdown of the untether protein TBC1D15 resulted in increased Ca$^{2+}$ transient (Fig. 4q, r). Of note, U18666A treatment, selectively impinged on short-range lysosome-mitochondria contacts leaving unaffected the long-range interactions probably indicative of a sterol-transfer mediated by very close contact at the basis of the observed phenotype and further strengthening the ability of the SPLICS reporters in specifically labeling short and long contact sites. Our in vivo data also demonstrated that our SPLICS

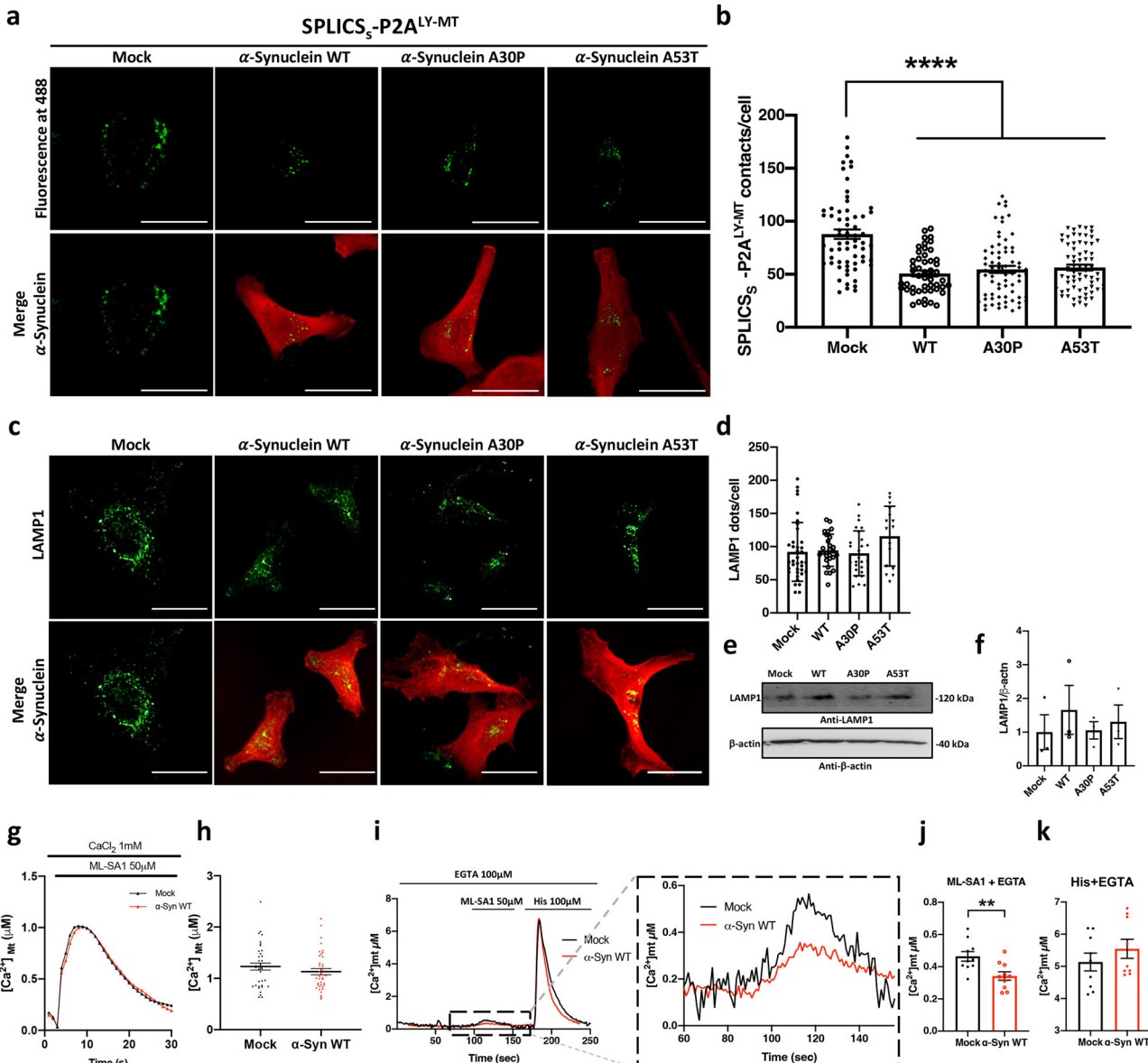

**Fig. 6 | Effects of α-synuclein overexpression on lysosomes–mitochondria contacts. a** Representative Z-projection images of HeLa cells expressing SPLICS$_S$-P2A$^{LY-MT}$ alone (Mock) or co-expressing SPLICS$_S$- P2A$^{LY-MT}$ and α-Syn WT or α-Syn A30P or α-Syn A53T. SPLICS$_S$- P2A$^{LY-MT}$ (488 nm), α-synuclein detected by anti- α-synuclein (633 nm) is visible in the merge panels. **b** Quantification of short LY-MT contacts (Mean ± SEM: SPLICS$_S$- P2A$^{LY-MT}$ + Mock 87.84 ± 4.46 n = 64; SPLICS$_S$- P2A$^{LY-MT}$ + WT 50.67 ± 2.77 n = 50; SPLICS$_S$- P2A$^{LY-MT}$ + A30P 54.60 ± 3.39 n = 70; SPLICS$_S$-P2A$^{LY-MT}$ + A53T 56.35 ± 2.69 n = 69). **c** Representative Z-projection images of HeLa cells transfected with pcDNA3 (Mock) or α-Syn WT or α-Syn A30P or α-Syn A53T. Lysosomes were stained by anti-LAMP1 (488 nm) and α-synuclein was detected by anti- α-synuclein (633 nm) and visible in the merge panels. **d** Quantification of LAMP1 dots per cell. The SPLICS dots and LAMP1 signal were quantified from the 3D rendering of a complete Z-stack, mean ± SEM (Mock 92.04 ± 7.27 n = 37; WT 94.41 ± 4.66 n = 27; A30P 89.92 ± 6.63 n = 26; A53T 115.7 ± 11.29 n = 16). **e, f** Western blotting analysis and quantification of LAMP1 expression in Mock, or α-Syn WT or α-Syn A30P or α-Syn A53T transfected cells. Equal amount of total loaded lysate was

verified by incubation with anti-β actin antibody (Mock 1 ± 0.51, WT 1.66 ± 0.73, A30P 1.05 ± 0.26, A53T 1.31 ± 0.50), data were obtained from 3 independent experiments. **g, h** Mitochondrial Ca$^{2+}$ transients generated upon ML-SA1 stimulation in HeLa cells expressing mtAeqwt alone (Mock) or co-expressing mtAeqwt and α-Syn WT and average peak values, mean ± SEM: [Ca$^{2+}$]$_{mt}$, μM Mock 1.230 ± 0.069 n = 37; α-syn WT 1.130 ± 0.059 n = 39. **i** Mitochondrial Ca$^{2+}$ transients in HeLa cells expressing mtAeqwt alone (Mock) or co-expressing mtAeqwt and α-Syn WT. The cells were first stimulated with ML-SA1 in a Ca$^{2+}$ free KRB supplemented with 100 μM EGTA to release Ca$^{2+}$ from lysosomes and then with histamine to release Ca$^{2+}$ from the ER. The inset shows the Ca$^{2+}$ transients induced by ML-SA1 application at a higher magnitude. **j** Average peak values upon ML-SA1 (mean ± SEM: [Ca$^{2+}$]$_{mt}$, μM Mock 0.46 ± 0.03 n = 10; α-syn WT 0.34 ± 0.03 n = 10) or (**k**) histamine stimulation were graphed as mean ± SEM: [Ca$^{2+}$]$_{mt}$, μM Mock 5.13 ± 0.83 n = 9; α-syn WT 5.55 ± 0.28 n = 9. Scale bar 10 μm. n = cells (**b, d**) or coverslips (**h, j, k**) examined over 3 independent experiments. **p ≤ 0.01 unpaired two-tailed t test, ****p ≤ 0.0001 one-way ANOVA. Source data are provided as a Source Data file.

reporter can be used in living Zebrafish and *Drosophila* models to study the role and the function of the lysosome-mitochondria interface in both physiological and pathological relevant context. Lastly, the availability of a reporter to monitor mitochondrial-lysosomes tethering allowed us to identify the PD-related protein α-synuclein as a player in the regulation of this interface and to show that the re-

organization of lysosome-mitochondria interface represents an early response to increased α-synuclein expression which in turn results in the calcium-dependent activation of TFEB. In agreement with our findings, TFEB nuclear translocation via calcineurin- and Rag GTPase-dependent pathways by TRPML1-mediated lysosomal Ca$^{2+}$ release[63,86,87] and early TFEB nuclear translocation upon α-syn overexpression in

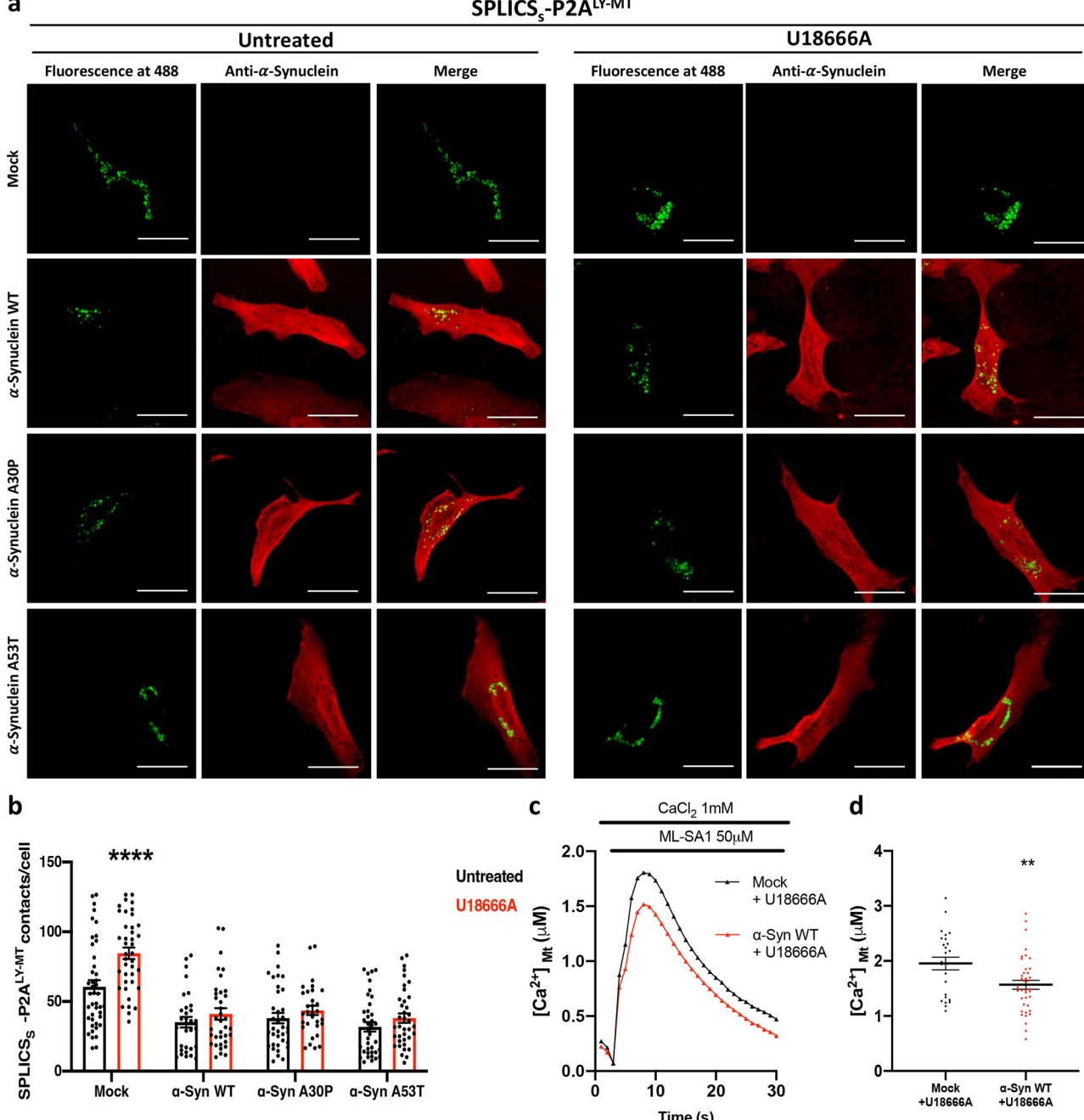

**Fig. 7 | Effects of α-synuclein overexpression on LY-MT contacts under cholesterol accumulation. a** Representative Z-projection images of HeLa cells transfected with SPLICS_S- P2A^{LY-MT} (Mock) or co-transfected with SPLICS_S- P2A^{LY-}MT and α-Syn WT or α-Syn A30P or α-Syn A53T in untreated condition or under U18666A treatment. SPLICS_S- P2A^{LY-MT} was represented by fluorescent "dots" upon excitation at 488 nm (green) and α-synuclein was detected by anti- α-synuclein upon excitation at 633 nm (red) and shown in the merge panels. **b** Quantification of short LY-MT contacts. The SPLICS_S- P2A^{LY-}MT contacts were quantified from the 3D rendering of a complete Z-stack, mean ± SEM Untreated: SPLICS_S- P2A^{LY-MT} + Mock 60.57 ± 4.73 $n = 43$, SPLICS_S- P2A^{LY-MT} + WT 35.15 ± 3.83 $n = 31$; SPLICS_S- P2A^{LY-}

MT + A30P 38.07 ± 3.78 $n = 36$; SPLICS_S- P2A^{LY-MT} + A53T 31.83 ± 3.39 $n = 38$. U18666A treated: SPLICS_S- P2A^{LY-MT} + Mock 83.49 ± 4.09 $n = 39$, SPLICS_S- P2A^{LY-MT} + WT 41.06 ± 4.09 $n = 36$; SPLICS_S- P2A^{LY-MT} + A30P 43.58 ± 3.30 $n = 31$; SPLICS_S- P2A^{LY-MT} + A53T 38.08 ± 3.37 $n = 38$) **c, d** Mitochondrial Ca^{2+} transients generated upon ML-SA1 stimulation in HeLa cells treated with U18666A and cotransfected with mtAeqwt and pcDNA3 (Mock) or α-Syn WT (mean ± SEM: [Ca^{2+}]_{mt}, μM Mock 1.954 ± 0.116 $n = 25$; α-syn WT 1.567 ± 0.082 $n = 39$). Scale bar 10 μm. $n$ = cells (**b**) or coverslips (**d**) examined over 3 independent experiments. (****$p ≤ 0.0001$ two-way ANOVA) (**$p ≤ 0.01$, unpaired two-tailed $t$ test). Source data are provided as a Source Data file.

dopaminergic neurons[52] have been previously reported, but no explanation for the signal "used" by α-syn to induce this response. We are aware that our data were obtained in HeLa cells, which do not represent a very relevant model to study α-syn- induced pathology, but the strong inhibitory effect of α-syn WT and its PD-related A30P and A53T mutants on lysosome-mitochondria tethering, and on Ca^{2+}

transfer into mitochondria is very interesting. All in all, our data support a model which proposes a role for α-syn as modulator of the lysosome-mitochondria interface and lysosomes-to-mitochondria Ca^{2+} transfer. Reduced mitochondrial Ca^{2+} uptake due the removal of lysosome-mitochondria tethering will result in perturbations of Ca^{2+}-microdomains near lysosomes with consequent calcineurin activation

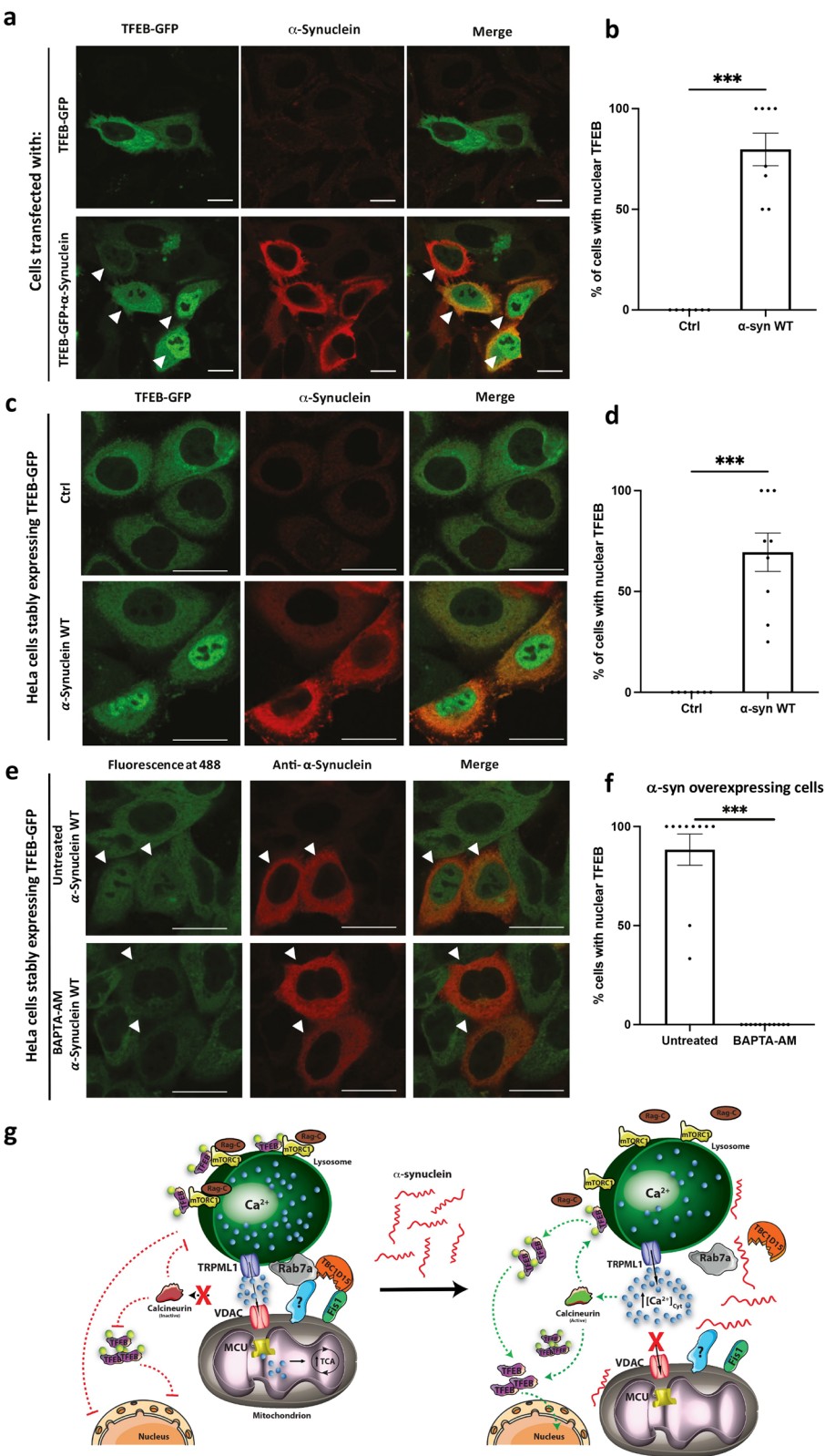

and TFEB translocation according to the model we propose in Fig. 8g). This action could be an early signal used by the cells to activate the response of protective autophagic pathway and counteract further accumulation of α-syn, that however, when excessive turns out in cytosolic TFEB sequestration and impinges on authophagic response, as proposed by Decressac et al.[52]. Overall, our study reports the generation and experimental validation of a genetically encoded SPLICS

probe to be used for in vitro and in vivo applications. It unravels the existence of at least two types of contact sites between mitochondria and lysosomes that are functionally distinct. Now, the challenge is to understand the mechanisms by which they are regulated and the physiopathological implications of their modulation. It will also be important to explore their role in vivo as well as to set up high throughput screening of compounds and/or players at the interface in

**Fig. 8 | TFEB subcellular localization under α-synuclein overexpression.**
**a** Representative single-plane images of HeLa cells expressing TFEB-GFP or co-expressing TFEB-GFP and α-Syn WT. **b** Quantification of nuclear TFEB-GFP intensity signal, mean ± SEM: Ctrl $0 ± 0$ $n = 7$, TFEB-GFP $79.76 ± 8.07$ $n = 8$ cells from 2 independent experiments. **c** Representative single-plane images of HeLa cells stably expressing TFEB-GFP and transiently transfected with pcDNA3 (Mock) or α-Syn WT. **d** Quantification of nuclear TFEB-GFP intensity signal. TFEB was represented by GFP fluorescence upon excitation at 488 nm (green) and α-synuclein was detected by anti-α-synuclein upon excitation at 633 nm (red); mean ± SEM: Ctrl $0 ± 0$ $n = 7$, TFEB-GFP $69.44 ± 9.52$ $n = 9$ cells from 2 independent experiments.
**e** Representative confocal images of HeLa cells stably expressing TFEB-GFP, transfected with α-Syn WT and left untreated or treated with BAPTA-AM (20μM,

6 h), quantification of the nuclear TFEB signal is shown in (**f**) mean ± SEM: Untreated $88.33 ± 7.86$ $n = 10$, BAPTA-AM $0 ± 0$ $n = 11$ cells from 3 independent experiments. **g** Proposed model: under basal conditions $Ca^{2+}$ released from lysosomes is taken up by mitochondria in close proximity, thus reducing calcineurin activation and TFEB nuclear translocation. The untethering of the Ly-Mt contacts by α-synuclein potentially increases the local $Ca^{2+}$ concentration due to reduced buffering action by mitochondria, thus leading to increased local activation of calcineurin- and Rag GTPase-dependent pathways and consequent TFEB nuclear translocation. Scale bar 10 μm. The data were obtained from at least 2 independent transfections.
***$p ≤ 0.01$, unpaired two-tailed $t$ test). Source data are provided as a Source Data file.

order to open avenues to prevent or delay debilitating diseases by targeting this local pathway.

# Methods

## Ethical statement

The research complies with all relevant ethical regulations. All experiments on mice were conducted according to the Italian Ministry of Health and the approval by the Ethical Committee of the University of Padova (Protocol Permit #690/2020-PR). No ethical committee is required for *D.rerio* experiments at 24hpf (University of Padova and Italian Ministry of Education and Research) and for *D. melanogaster* experiments at Kings College (not classified as protected species under the UK 'Animals Scientific Procedures Act (1986)').

## Cloning and fusion plasmid construction

To generate $SPLICS_S$-$P2A^{LY-MT}$, simultaneously expressing (through a 2A peptide) lysosomal β11 short ($Ly_S$- β11) and outer mitochondrial membrane $GFP_{1-10}$ ($OMM$-$GFP_{1-10}$), the coding sequence of humanized TMEM192 was PCR amplified from pLJC5-Tmem192-2xFlag, kindly provided by Dr. Nuno Raimundo, Institute of Cellular Biochemistry, Universitätsmedizin Göttingen, by using the primers For. (GGATCCA CCATGGCGGCGGGGGGCAGGATGGAGGACGGTTCCTTGG) and Rev. (GCTGTCAGCCAAGTAGAACGGGCGACGGCGGCAGCGGCGGCGGCA GCGGCCGGGACCACATGGTGCTGCACGAGTACGTGAACGCCGCTGG CATCACAGGCCCCGGG). The PCR product was purified using the GenElute Gel Extraction Kit (Sigma), digested with BamHI and SmaI and then ligated into the pcDNA3-$SPLICS_S$-$P2A^{ER-MT}$ vector digested with the same restriction enzymes[35,36]. Lysosomal β11 long ($Ly_L$- β11) were designed and obtained by custom gene synthesis (Vectorbuilder). $Ly_L$- β11 was insert into the pcDNA3-$SPLICS_S$-$P2A^{LY-MT}$ vector digested with BamHI and SmaI enzymes and then ligated. mCh-Rab7A (mCherry- Rab7A) (Plasmid #61804), pEF6-myc-TBC1D15 (c-myc-TBC1D15) (Plasmid #79148) were purchased from Addgene. mCh-Rab7A plasmid was mutagenized to obtain mCh-Rab7A(Q67L) and mCh-Rab7A(T22N) using QuikChange® XL mutagenesis kit (For. ACTGGAACCGTTCCAGTCCTGCTGTGTCC and Rev. GGACACAGCAGG ACTGGAACGGTTCCAGT for mCh-Rab7A(Q67L), (For. CTGGTTCATGA GTGAGTTCTTCCCGACTCCAGAATCTCC and Rev. GGAGATTCTGG AGTCGGGAAGAACTCACTCATGAACCAG for mCh-Rab7A(T22N)). pEF6-myc-TBC1D15 was mutagenized to obtain pEF6-myc- TBC1D15 (D397A) using QuikChange® XL mutagenesis kit (For. CGATCTG TTCTGTTGACAGCTTTCTCGATCAGACTTC and Rev. GAAGTCT-GATCGAGAAAGCTGTCAACAGAACAGATCG). Cytosolic and mitochondrially targeted aequorin constructs (cytAEQ and mtAEQwt) and α-synuclein WT, A53T and A30P constructs are described elsewere[71,89]. To knockdown Rab7A or TBC1D15 two different siRNAs against Rab7A or TBC1D15 were used (Silencer Pre-designed siRNA Ambion Cat#: AM16708; Gene: Rab7A, Clone #1 ID:139428 and Clone #2 ID:139430 and Gene: TBC1D15, Clone #1 ID:148837 and Clone #2 ID:148839) (ThermoFisher Scientific). pEGFP-TFEB was kindly provided by Dr. Gennaro Napolitano, Telethon Institute of Genetics and Medicine (TIGEM), Naples, Italy.

## Cell Culture

HeLa ATCC cells and HeLa cells stably expressing TFEB-GFP, were grown at 37 °C in a 5% $CO_2$ atmosphere in Dulbecco's modified Eagle's medium (DMEM) high glucose (Gibco; 41966-029), supplemented with 10% Fetal Bovine Serum (FBS) (Gibco; 10270-106), 100 μ/ml penicillin and 100 mg/ml streptomycin (Penicillin–Streptomycin solution 100X) (EuroClone; ECB3001D). All treatments were performed after transfection; particularly, U18666A (Sigma-Aldrich; 662015) treatment was performed for eighteen hours with 2 μg/ml, vehicle (water) was added in untreated cells as control. Starvation experiments were performed incubating HeLa cells for three hours in Hanks' Balanced Salt Solution (HBSS) (Gibco; 14025-050) or adding in the grown medium 1 μM rapamycin (Sigma-Aldrich; 37094) for five hours. CCCP (Sigma-Aldrich; C2759) treatment was performed for three hours with 10 μM while oligomycin/antimycin A (Sigma-Aldrich; A8674) were both used at 10 μM for one-half-hour. Vehicle (DMSO) was added in untreated cells as control. Primary mouse cortical neurons were isolated and transfected as previously reported[90]. Primary mouse cortical neuronal cultures were obtained from post-natal day 0 pups by papain dissociation of the brain tissue and maintained as previously described[90]. At days-in-vitro 4, neurons plated in 24-well plates were transfected with lipofectamine (Lipofectamine 2000, Invitrogen) in a 1:2 ratio with DNA (v/w). The transfection was carried in Opti-MEM (GIBCO Life Technologies) for 45 min. Transfected neurons were then maintained in culture for additional 3–5 days prior to fixation. Four independent cultures and transfection were performed.

## Transfection

Twenty hours before transfection, HeLa cells were seeded at 40–50% confluence onto 13 mm diameter glass coverslips (for immunofluorescence analysis) or 60–70% in 6-well plate (for Western blotting analysis). HeLa cells were transfected by standard $Ca^{2+}$ phosphate protocol that was previously reported[71]. Briefly, for one 13 mm coverslip, 5 μl of 2.5 M $CaCl_2$ (Sigma-Aldrich; Cat# C-5080) was added to 2 μg of total $SPLICS_{S/L}$-$P2A^{LY-MT}$ DNA dissolved in Milli-Q $H_2O$ to reach a final volume of 50 μl. $CaCl_2$-DNA solution was added drop by drop to 50 μl HEPES Buffered Solution 2X (HBS 2X: 280 mM NaCl, 50 mM Hepes, 1.5 mM $Na_2HPO_4$·$7H_2O$ (Sigma-Aldrich; Cat# S9390), pH 7.12) and incubated 30 min at room temperature. Before transfection, the growth medium was replaced with fresh medium. Eight hours after transfection cells were washed three times with Dulbecco's Phosphate Buffered Saline (D-PBS) (EuroClone; Cat# ECB4004L) to remove excess of $Ca^{2+}$ phosphate precipitates and fresh medium was replaced, incubating cells for additional thirty-six hours. In co-transfection experiments, $SPLICS_{S/L}$-$P2A^{LY-MT}$ were co-transfected in a 2:1 ratio for mCh-Rab7AWT, mCh-Rab7A(Q67L), mCh-Rab7A(T22N), pEF6-myc-TBC1D15WT and pEF6-myc-TBC1D15(D397A) while in a 1:2 ratio for α-synuclein WT, A53T and A30P plasmids. Mock cells were transfected with $SPLICS_{S/L}$-$P2A^{LY-MT}$ and empty vector in the same respectively conditions. To knockdown Rab7A or TBC1D15, HeLa cells were transfected using Lipofectamine 3000 Transfection Reagent (Invitrogen, Cat. L3000001) in accordance with the manufacturer's instructions.

Briefly, HeLa cells were seeded in a 6-well plate at 70–90% of confluence and transfected with Lipofectamine 3000 Transfection Reagent plus 1.5 μg SPLICS$_{S/L}$-P2A$^{LY-MT}$ and 60 pmol of different siRNAs. A scramble siRNA was used as a control. Thirty-six-hours after transfection cells were washed and re-plated at 40–50% confluence onto 13 mm diameter glass coverslips (for immunofluorescence analysis) or 60–70% in 12-well plates (for Western blotting analysis) for additional twenty-four-hours. EGFP-TFEB and α-synuclein WT plasmids were transfected with FuGENE® Transfection Reagent (Promega; Cat# E2311) in accordance with the manufacturer's instructions. Briefly, HeLa cells were seeded at 50–60 % of confluence onto 13 mm diameter glass coverslips twenty hours before transfection. For one 13 mm coverslip, cells were transfected with FuGENE Transfection Reagent plus 25 ng of EGFP-TFEB plasmid and 300 ng of α-synuclein WT plasmid and incubated 30 min at room temperature. Mock cells were transfected with EGFP-TFEB or α-synuclein WT plasmid and pcDNA3(+) empty vector in the same respectively conditions. Before transfection, the growth medium was replaced with fresh medium.

## Immunocytochemistry

Forty-eight hours post transfection cells, plated on 13- mm glass coverslips, were fixed for 20 min in a 3.7% (vol/vol) formaldehyde solution (Sigma- Aldrich; Cat# F8775). Cells were then washed three times with D-PBS (Euroclone). Cell permeabilization was performed by 10 min incubation in 0.3% Triton X-100 Bio-Chemica (PanReac AppliChem; A1388) in D-PBS, followed by three times washes in 1% gelatin/D-PBS (Type B from bovine skin) (Sigma-Aldrich; G9382) for 15 min at room temperature (RT). The coverslips were then incubated for 90 min at RT with the specific primary antibody diluted in D-PBS (1:50 anti-GFP: Santa Cruz, Cat#sc_9996, 1:50 Tom20: Santa Cruz. Cat#sc-11415, 1:20 anti-LAMP1: Santa Cruz, Cat#sc_20011, 1:50 anti-α-Syn: Santa Cruz, Cat#sc_12767, 1:100 c-Myc: Cell Signaling (D84C12) Cat#5605). Three washing with 1% gelatine/D-PBS were performed to remove the excess of primary antibody. Staining was revealed by the incubation with a dilution 1:100-1:200 in D-PBS of specific Alexa Fluor secondary antibodies (Thermo Fisher: Goat anti-Rabbit IgG Alexa Fluor 405, A-31556; Goat anti-Rabbit IgG Alexa Fluor 594, A-11012; Goat anti-Mouse IgG Alexa Fluor 405, A-11012; Goat anti-Mouse IgG Alexa Fluor 633, A-31553, Goat anti-Rabbit IgG Alexa Fluor 488, A-11034) for 45 min at room temperature. After three additional washing with 1% gelatine/D-PBS, coverslips were mounted using Mowiol 40–88 (Sigma-Aldrich; 81386). Mitochondria morphology of transfected HeLa was detected by MitoTracker™ Red CMXRos (Invitrogen, Cat. M7512) in accordance with the manufacturer's instructions. Briefly, before fixing, cells were washed in Hank's Balanced Salt Solution (HBSS; GIBCO) and incubated with 150 nM MitoTracker ® Red CMXRos (Invitrogen, Cat. 9082), for 15 minutes at 37 °C in a 5% CO2 atmosphere. Cholesterol accumulation under U18666A treatment was detected by Filipin III staining (Sigma-Aldrich, Cat. SAE0087) following the manufactures instructions. Briefly, after treatment, cells were fixed 2 h in 4% paraformaldehyde PH 7.4 (Sigma-Aldrich, Cat. 158127) and incubated 2 h in 0.05 mg/ml of Filipin III solution. After three additional washing with D-PBS, coverslips were mounted using Mowiol 40–88.

## Confocal microscopy and image analysis

Cells were imaged with a Leica TSC SP5 inverted confocal microscope, or Zeiss LSM700 using either a HCX PL APO 100X/numerical aperture 1.40–0.60 or a HCX PL APO × 100/numerical aperture 1.4 oil-immersion objective. Images were acquired by using the Zeiss or Leica AS software. The images were acquired at lasers wavelength of 405, 488, 555 and 633 nm. Z-stack of the cells were acquired every 0.29 μm, then processed using ImageJ (National Institutes of Health (NIH)). Images were first convolved, and the cells were selected by freehand selection of ImageJ in the drawing/selection polygon tool and then processed using the "Quantification 1" plugin (https://github.

com/titocali1/Quantification-Plugins). A 3D reconstruction of the resulting image was obtained using the VolumeJ plugin (https://github. com/titocali1/Quantification-Plugins). A selected face of the 3D rendering was then thresholded and used to count short and long contact sites through the "Quantification 2" plugin (https://github.com/titocali1/Quantification-Plugins). Radial analysis was performed as described in[91] by using the Radial_Profile.class plugin of ImageJ software (https://imagej.net/ij/plugins/radial-profile.html).

## Western blotting

Transfected cells were recovered from 6-well plate and cellular extracts were western blotting to evaluate GFP, TMEM192, LAMP1, Rab7A or TBC1D15 proteins expression. Cells were lysed in RIPA Buffer (50 mM Tris-HCl pH 7.4, NaCl 150 mM, 1% Triton X-100, 0.5% sodium deoxycholate, 10 mM EDTA, 0.1% SDS, 1 mM DTT, 2X Protease Inhibitor Cocktail (Sigma, Cat# P8340)) for 20 min. Supernatants were collected after 20 min centrifugation at $16,000 \times g$ at 4 °C. Extracted proteins were quantified by Bradford assay (Bio-Rad, Cat# 500-0205), resolved on SDS-PAGE in 12% SDS/PAGE Tris- HCl polyacrylamide gel, and then transferred to PVDF membranes (BioRad) using Trans-Blot® Turbo™ Transfer System (BioRad). Membranes were blocked with 5% w/v non-fat dried milk (NFDM) in TBST (20 mM Tris-HCl, pH 7.4, 150 mM NaCl, 0.05% Tween-20) and incubated overnight with the specific primary antibody at 4 °C. Signal was detected by incubation with secondary horseradish peroxidase-conjugated anti-rabbit (Santa Cruz, sc-2004) or anti-mouse (Santa Cruz, sc-2005) IgG antibodies for 1 h at room temperature followed by incubation with the chemiluminescent reagent Luminata Classico HRP substrate (Merck Millipore, Cat# WBLUO500). The following primary antibody were used: monoclonal anti-GFP 1:1000 (Santa Cruz, Cat. Sc-9996), monoclonal anti-TMEM192 1:1000 (Abcam, Cat. ab186737), polyclonal anti-tubulin 1:2000 (Cell Signaling, Cat. #2146), monoclonal anti-LAMP1 (Santa Cruz, Cat. Sc-20011), monoclonal anti-beta actin (Sigma-Aldrich, Cat. A536), polyclonal anti-Rab7A 1:500 (Sigma-Aldrich, Cat. HPA006964), polyclonal anti-TBC1D15 1:500 (Sigma-Aldrich, Cat. HPA013388), monoclonal anti-GAPDH 1:2000 (Cell Signaling, Cat. 2128), monoclonal anti-vinculin 1:5000 (Sigma-Aldrich, Cat. V9264).

## Aequorin measurements

For Ca$^{2+}$ measurements, cells were co-transfected with low-affinity mitochondrial aequorin (mtAeqmut) or high-affinity mitochondrial aequorin (mtAeqwt) or cytosolic aequorin (cytAEQ) and SPLICS$_{S/L}$-P2ALY-MT or α-synuclein WT plasmid following standard Ca$^{2+}$ phosphate protocol. Briefly, cells were seeded at 40–50% confluence onto 13 mm diameter glass coverslips and twenty hours later cells were transfected adding 5 μl of 2.5 M CaCl2 (Sigma-Aldrich; Cat# C-5080) to 2.5 μg of total DNA (2:1 ratio favoring pcDNA3.1(+) empty vector or SPLICS$_{S/L}$-P2ALY−MT/ α-synuclein WT protein expressing plasmids) dissolved in Milli-Q H$_2$O to reach a final volume of 50 μl of HBS 2X. Eight hours after transfection cells were washed three times with Dulbecco's Phosphate Buffered Saline (D-PBS) (EuroClone; Cat# ECB4004L) to remove excess of Ca$^{2+}$ phosphate precipitates and fresh medium was replaced for additional thirty six-hours. Forty-eight hours post-transfection, mitochondrial high affinity aequorin (mtAeqwt) was reconstituted by incubating cells for 1 h with 5 μM WT coelenterazine (Invitrogen; C2944) in KRB supplemented with 0.1% glucose and 1 mM CaCl$_2$ (Sigma-Aldrich; 21115) at 37 °C in a 5% CO$_2$ atmosphere. After reconstitution, cells were transferred to the chamber of luminometer, and Ca$^{2+}$ transient were mensurated by perfusion in KRB medium added with 1 mM CaCl$_2$ and 50 nM ML-SA1 (Sigma-Aldrich; SML0627-25MG) for SPLICS$_{S/L}$-P2ALY−MT transfected cells or 100 μM EGTA (Sigma-Aldrich; E-4378), 1 mM CaCl$_2$, 100 μM Histamine, 50 nM ML-SA1 (Sigma-Aldrich; SML0627-25MG) for α-synuclein WT transfected cells. Finally, cells were lysed with 100 μM digitonin in a hypotonic CaCl$_2$ rich solution (10 mM CaCl$_2$ in H$_2$O) to discharge the remaining

reconstituted active aequorin pool. For cytosolic and mitochondria $Ca^{2+}$ measurements under U18666A treatment or in presence of mCh-Rab7AWT or mCh-Rab7A(Q67L) or mCh-Rab7A(T22N) or pEF6-myc-TBC1D15WT or pEF6-myc-TBC1D15(D397A) or α-synuclein WT, cells were co-transfected with cytosolic aequorin (cytAEQ) or wt mitochondrial aequorin (mtAeqwt) and pcDNA3.1(+) empty vector or protein expressing plasmids in a 1:2 ratio using Lipofectamine 2000 Transfection Reagent (Invitrogen, Cat. 11668019) in accordance with the manufacturer's instructions. Briefly, HeLa cells were seeded in a 6-well plate at 70–90% of confluence and transfected with Lipofectamine 2000 Transfection Reagent plus 1 μg mtAeqwt or cytAEQ and 2 μg of different DNA plasmids except for α-synuclein WT where 1 μg was used. pcDNA3.1(+) empty vector was used as a control. Six hours after transfection cells were washed and were re-plated into a 96-well plate (Corning®; Cat# 3610) (for $Ca^{2+}$ analysis) or in 12-well plates (for Western blotting analysis). For $Ca^{2+}$ measurements in presence of Rab7A or TBC1D15 siRNAs, cells were transfected using Lipofectamine 3000 Transfection Reagent (Invitrogen, Cat. L3000001) in accordance with the manufacturer's instructions. Briefly, HeLa cells were seeded in a 6-well plate at 70–90% of confluence and transfected with Lipofectamine 3000 Transfection Reagent plus 1 μg mtAeqwt and 60 pmol of different siRNAs. A scramble siRNA was used as a control. Thirty-six hours after transfection cells were washed and re-plated into a 96-well plate (Corning®; Cat# 3610) (for $Ca^{2+}$ analysis) or in 12-well plates (for Western blotting analysis). Thirty-six hours post re-plated, aequorins were reconstituted by incubating cells for 90 min with 5 μM coelenterazine (Santa Cruz Biotech.; Cat# sc-205904) in KRB solution supplemented with 0.1% glucose and 1 mM $CaCl_2$ at 37 °C. Luminescence measurements were carried out using a PerkinElmer EnVision plate reader equipped with two injector units. For mitochondrial and cytosolic $Ca^{2+}$ measurements, after reconstitution, cells were placed in 40 μl of KRB solution with 0.1% glucose and 1 mM $CaCl_2$ and luminescence from each well was measured for 30–40 s. During the experiment, 50 nM ML-SA1 at the final concentration was first injected to activate $Ca^{2+}$ transients, and then a hypotonic, $Ca^{2+}$-rich, digitonin (Sigma- Aldrich; Cat# D5628) containing solution was added to discharge the remaining aequorin pool. Output data were analyzed and calibrated with a custom-made macro-enabled Excel workbook.

## Zebrafish husbandry and imaging

All animal experiments were conducted on wild-type fish. Adult fish were maintained and raised in 5 l tanks with freshwater at 28 °C with a 12 h light/12 h dark cycle. Embryos were obtained from spontaneous spawnings and raised at 28 °C in Petri dishes containing fish water[92]. To perform experiments, both wt and s1102t:GAL4 fish were used. All experiments were conducted on 24 h post fertilization (hpf) embryos. The pT2-DsRed-UAS-SPLICS$_S$-P2A vector has been already described[35]. Before injections, all plasmids were diluted in Danieau solution (58 mM NaCl, 0.7 mM KCl, 0.4 mM $MgSO_4$, 0.6 mM $Ca(NO_3)_2$, 5 mM HEPES pH 7.6) and 0.5% phenol red. At 24 hpf, embryos were screened for fluorescence, dechorionated, and anesthetised with tricaine. They were then mounted on 35 × 10 mm glass bottom Petri dishes (Ted Pella, INC. Prod. No 14023-20) in low melting agarose (1.3%, Euro-Clone). Fish water containing tricaine methanesulfonate 0.61 mM (Sigma) was added in the Petri dishes, in order to keep fish anesthetised. Mounted fish were imaged at RT (20–23 °C) using a Leica TSC SP5 inverted confocal microscope, using a HCX PL APO 63X/numerical aperture 1.40-0.60 oil-immersion objective. To image ly–mt contacts, a Z-stack of the cell was acquired.

## Drosophila husbandry, generation of transgenic flies and Imaging

Flies were maintained on "Iberian" food [70 mg/ml yeast (Brewer's yeast, MP Biomedicals, 903312), 55 mg/ml glucose (VWR, 10117HV),

7.7 mg/ml agar (SLS, FLY1020), 35 mg/ml organic plain white flour (Doves Farm, UK), 1.2 mg/ml Nipagin (Sigma, H3647), 0.4% propionic acid (Sigma-Aldrich, P5561] at 25 °C and 60% humidity with a 12-h light–12-h dark cycle. The transgenic *UAS-SPLICS$_S$-P2A$^{LY-MT}$* fly lines generated in this study were obtained by phiC31-mediated transgenesis to integrate the relevant constructs into either the attP40 (25C6) or attP2 (68A4) landing sites following embryo injection. Imaging was performed on a spinning disk confocal microscope as previously reported[93] and quantification of SPLICS puncta was performed in the neurons of the wing margins from single focal planes.

## Statistics and reproducibility

Results are reported as means ± SEM and Gaussian distribution was assessed by D'Agostino-Pearson omnibus and Shapiro-Wilk normality tests. Statistical analysis of two groups was obtained applying unpaired Student's two-tailed $t$ test. To compare more of two groups one-way/two-way ANOVA test was used. All statistical analyses were performed using GraphPad Prism version 8.00 for Mac OS X (La Jolla, California, USA). Statistical significance threshold was set at $p < 0.05$. The values of $n$ is indicated in the figure legends. *$p \leq 0.05$, **$p \leq 0.01$, ***$p \leq 0.001$, ****$p \leq 0.0001$.

## Reporting summary

Further information on research design is available in the Nature Portfolio Reporting Summary linked to this article.

## Data availability

The data that support this study are present in the manuscript and supplementary information, and are available from the corresponding authors upon request. Requests will be fulfilled within 4 weeks. Source data are provided with this paper.

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

## Acknowledgements

The work is supported by grants from the Ministry of University and Research (PRIN2017 no 2017E5L5P3 to T.C. and PRIN2022 no 20223ABZ82 to M.B.), from the Università degli Studi di Padova (STARS Consolidator Grant 2019 to T.C., Progetto di Ateneo 2023 no. CALI_BIRD23_01 to T.C.), PNRR – CN3 National Center for Gene Therapy and Drugs based on RNA Technology to M.B. and T.C. n. CN00000041 (2022-26), funded by the European Union-Next Generation EU; NC3Rs SKT grant (NC/T001224/1 to A.V.) and a King's Together Multi and Interdisciplinary Research Scheme (Wellcome Trust Institutional Strategic Support Fund (grant reference: 204823/Z/16/Z) to A.V). G.N. is supported by the Italian Telethon Foundation (TGM22CBDM09); Associazione Italiana per la Ricerca sul Cancro A.I.R.C. (MFAG-23538); MIUR (PRIN 2017YF9FBS); Wereld Kanker Onderzoek Fonds (WKOF), as part of the World Cancer Research Fund International grant program (IIG_FULL_2022_009). We thank the Wohl Cellular Imaging Centre at King's College London for help with light microscopy, the Fly Facility of the Department of Genetics, University of Cambridge for help with *Drosophila* embryo injections, and the Zebrafish Facility of the Department of Biology, University of Padova.

## Author contributions

F.G., L.B., F.M., E.P., A.M., A.V., T.C. and M.B. contributed to the data curation and methodology; F.G., T.C. and M.B. contributed to the conceptualization and writing-original draft preparation; F.G., L.B., E.P., G.N., A.V., T.C. and M.B. contributed to revising and editing original draft preparation; A.E. and G.N. contributed to TFEB experiments, T.C. and M.B. contributed to funding acquisition.

## Competing interests

The authors declare no competing interests
