## [Peer Review File · Nature Communications]

REVIEWER COMMENTS

Reviewer #1 (Remarks to the Author):

In their paper "A novel SPLICS reporter reveals α -synuclein mediated regulation of lysosomes-mitochondria contact sites and TFEB nuclear translocation", Giamogante and co-workers describe a genetically encoded SPLICS reporter engineered to measure short- and long- juxtapositions between mitochondria and lysosomes. The authors show that the overexpression of α -synuclein (α -syn) strongly reduced mitochondria/lysosomes contacts and affected the transfer of Ca^{2+} from lysosomes to mitochondria, which enhanced TFEB nuclear translocation. The study is well conducted and comprehensive, however some issues reduce the enthusiasm of this reviewer.

The authors have previously used a similar approach to determine ER-mitochondria contact sites (Vallese et al., 2020 doi: 10.1038/s41467-020-19892-6, Cali and Brini, 2021 doi: 10.1007/978-1-0716-1262-0_23). However, in the 2020 paper they used live imaging of ER-mitochondria contact sites in zebrafish RB neurons to prove their premise. In this work the authors only use HeLa cells, that are not the most convincing model to study mitochondria-lysosome contact sites in polarized cells like neurons, neither to extrapolate to neuronal demise in Parkinson's disease. So, the authors should validate SPLICSS/L-P2AMT-LY reporter in neurons and if possible in vivo.

In particular, the whole section dealing with α Synuclein effect on mitophagy and the correlations made by the authors need to be revised.

The authors show that α -syn WT, A30P and A53T drastically decreased the number of short contact sites compared to Mock cells, which is intriguing, since CCCP increased lysosome-mitochondria contacts to trigger mitophagy.

Several reports already shown that α Synuclein binds to mitochondrial membranes and induces its dysfunction and fragmentation, similar to CCCP. Moreover, Choi and colleagues (doi.org/10.1038/s41593-022-01140-3) showed that α Synuclein converts from a monomeric state into two distinct oligomeric states in neurons in a concentration-dependent and sequence-specific manner. This intracellular seeding events occur preferentially at mitochondrial membranes, leading to the impair complex I activity and increase mitochondrial reactive oxygen species (ROS) generation, which accelerates the oligomerization of A53T α -Syn and causes permeabilization of mitochondrial membranes and cell death. Nevertheless, authors say that an increase in α -Syn expression levels results in the activation of TFEB, possibly as a consequence of its action on lysosomes-mitochondria contact sites that leads to reduced Ca^{2+} buffering by mitochondria. Would this lead to an increase of the expression of lysosomal enzymes and LC3, so an increase in autophagy/mitophagy? However, the authors in HeLa cells expressing α -syn WT under HBSS treatment show no activation of macroautophagy!

Indeed, in 2013, Decressac and coworkers (doi: 10.1073/pnas.1305623110) showed a bi-temporal response of α Synuclein overexpression regarding the subcellular localization of the TFEB. They found that an early response to α -syn overexpression appeared to be associated with TFEB translocation into the nucleus, whereas at a later stage, deficiency in autophagy was linked to a reduced expression in the nuclear fraction due to the fact that α Synuclein binds to TFEB contributing to sequestration of this transcription factor into the cytoplasm, thus hampering the ALP response and, thereby, its own clearance. This data should be discussed in light of the authors new results.

Additionally, the role of mitochondria-lysosome contact sites in mitophagy in polarized cells, like neurons, is probably different than in HeLa cells. This should be at least discussed.

The paper has merit, although major revisions are required.

Reviewer #2 (Remarks to the Author):

Summary.

In this manuscript, Giamogante and colleagues designed and described, in a well-written manuscript, a couple of new probes to visualize short and long lysosome-mitochondria contacts

with conventional confocal microscopy. These probes were used to evaluate the impact of Rab7A GTPase and its GTPase-activating protein (TBC1D15) on the contacts between both organelles. In the same way, how these contacts were affected by different stressors and pathological conditions was studied with these probes. Once they observed this dynamic, calcium flux was evaluated and related to the contact differences with repercussions on TFEB localization. These experiments demonstrated that the probes designed by this group represent a powerful tool to evaluate the dynamic of lysosome-mitochondria contacts. However, there are several concerns that should be addressed.

Major points.

During the whole manuscript, the authors used transfection as DNA delivery and expression methodology for the probe, reporter, siRNA, and mutants. Since it is not clear how efficient transfection and co-transfection are in the experiments, the authors should explain the advantages of using transfection instead of generating stable cell lines with infection. In this way, there will be near to 100% cells with the probes, more co-expression with reporters and mutants would be reached, and the knockdown experiments would be more reliable.

In "Generation and characterization of SPLICSS/L-P2ALY-MT reporter for lysosomes mitochondria contact sites". The authors showed the design of the construct they used and observed that the signal generated by the reporter co-localized with lysosomes and mitochondria. To discard that this signal is a false positive, it is suggested to use both protein reporters separately. In the same way, they should mention whether this reporter works if the big part of GFP is linked to the mitochondrial protein and the β 11 strand to the lysosomal protein. The authors showed that the mitochondrial protein reporter 100% localizes with mitochondrial markers, but the same is not observed with the lysosomal protein reporter. The authors are advised to use the same anti-GFP antibody that was used to confirm the mitochondrial part.

In "Modulation of lysosomes-mitochondria contact sites during autophagy and mitophagy". The authors should work on finding the biological reasons why there is an increase in the contact number upon the stressors. What would happen if there were no more contacts? Is this increase in contact number Rab7A-dependent? These questions might be answered using the Rab7A T22N mutant. What happens if the lysosome activity is blocked with bafilomycin or chloroquine? More or less contacts? The authors are suggested to evaluate the contact number under these conditions. Are these contacts favoring mitophagy? A mitophagy reporter might be used to evaluate this upon CCCP or OA/AA treatment and with Rab7A mutants. Finally, the authors should explain the differences between HBSS and rapamycin treatment, both inhibit mTOR and increase autophagy but only rapamycin increases short contacts.

It is recommended to repeat the WB of Figure 5e to have a better representative image, more equal LAMP1 detection (big differences in the complete blot image are shown), and better quantification. In case the WB is not successful, the use of another lysosomal marker is recommended.

Since it is shown that extracellular calcium affects mitochondrial calcium uptake in Figure 5i, it is recommended to evaluate the participation of extracellular calcium in the increased mitochondrial calcium uptake modulated by Rab7A Q67L and TBC1D15 D39A or its knockdown.

In " α -Synuclein overexpression counteracts the increase of Ly-Mt short interactions induced by lysosomal cholesterol accumulation". The authors should prove evidence that lysosomal cholesterol is still increased with U18666A treatment and α -syn WT expression.

In " α -Synuclein overexpression counteracts Ly-Mt short interaction under starvation". There is no reason to use HBSS as a stressor here since in Figure 3 the authors showed that this stress does not affect the lysosome-mitochondria interactions. On the other hand, rapamycin, CCCP or OA/AA are better stimuli since both autophagy and Ly-Mt short interactions are increased.

In " α -Synuclein was involved in TFEB cellular localization". Western blot experiments are recommended to evaluate the phosphorylation of TFEB (anti-phospho TFEB) and other mTOR targets (S6K, S6 or 4E-BP1) to confirm that the mTOR-TFEB branch is inhibited but the canonical mTOR branch is not affected. The activity of TFEB is recommended to evaluate by qPCR of TFEB target genes. Since the authors suggest that the nuclear translocation of TFEB when α -syn is overexpressed is due to an increase in calcium localized in microdomains and calcineurin-dependent de-phosphorylation of TFEB, it is suggested to evaluate TFEB phosphorylation by WB and localization by IF or using GFP-TFEB construct when Calcineurin is knockdown and α -syn overexpressed.

Minor points.

Line 668 "t" to "T" after the period.

Reviewer #3 (Remarks to the Author):

In this manuscript, Giamogante et al describe and validate the development of a new probe that functions as a distance sensor between the membranes of mitochondria and lysosomes, and a reporter of contact between these two organelles in response to molecular manipulations and metabolic and proteostatic stresses. The experiments are well executed with appropriate controls and statistical analyses, and the data in general support the conclusions.

Fundamentally, the manuscript is centered on the development and validation of the SPLICS reporter probe. Results from the latter are used for further experimentation on the effects of α -synuclein on lysosome-mitochondria crosstalk, and lysosome-based function and signaling. This part of the manuscript is not very strong in terms of a conceptual advance, and especially for titling the manuscript after its findings. Technically, it is not accurate to claim that the SPLICS reporter reveals α -synuclein mediated regulation of TFEB nuclear translocation. The paper would be much stronger as a technical paper that reports and validates a new probe rather than over-extending incremental findings on α -synuclein, and making them a central theme of the paper. The α -synuclein data are also subject to the caveat that were conducted in HeLa cells, which is not a physiologically relevant cell system for studies of α -synuclein-triggered pathology. Moreover, the α -synuclein results seem to reproduce similar results reported by others in other systems. Therefore, they are confirmatory in nature and low in novelty.

Major Concerns:

- The short and long linkers are designed for reporting interactions between 4 and 10 nm distances. However, it is unclear whether these are bona fide organelle-organelle contacts. The manuscript does not provide any higher- and super-resolution images of juxtaposed mitochondria and lysosomal membranes at domains of the SPLICS signal.

The authors should show higher magnification zoomed-in images of lysosome and mitochondria membranes at sites of SPLICS signal (4 and 10 nm) – ideally obtained with super-resolution microscopy – for unequivocal demonstration of a correlation between the SPLICS signal and membrane-membrane contacts.

In the case of the 4 nm SPLICS signal, are mitochondria-lysosome contacts more extensive and tighter along the majority of the edge of each organelle in comparison to 7 nm, which are more loose and not as extensive laterally and not as closely apposed?

- A major concern with the Rab7 and starvation manipulations is the retrograde movement of lysosomes to perinuclear and juxtannuclear regions. In these regions, membrane density is very high, and organelle-organelle contacts might be the result of non-specific crowding. How can the authors rule out the latter possibility?

The authors should spatially distinguish between SPLICS signal occurring in the perinuclear and peripheral regions, and normalize it to the surface occupied by the organelle fluorescence signal. Does SPLICS signal per organelle surface area increase in peripheral non-perinuclear regions of the cytoplasm in response to starvation and Rab7 activation?

An advantage of the SPLICS probe/technology is that it can provide spatial information on the intracellular location of organelle-organelle contacts. This aspect has not been explored in the manuscript, and seems more fitting for the manuscript than the α -synuclein experiments and directions.

- As highlighted above, the α -synuclein experimentation is performed in HeLa cells rather than primary neurons. This is a major caveat that reduces confidence in the applicability of results in a physiological context. Moreover, the TFEB translocation in response to α -synuclein has been reported in neurons before, as the authors themselves cite. In addition, the experimental evidence

is not strong enough to unequivocally demonstrate that the enhanced TFEB translocations is a result of defective Ca^{++} transfer and release from lysosomes due to defective mitochondria-lysosome contacts. More is needed to demonstrate that intra-organelle Ca^{++} concentration is directly impacted by mitochondria-lysosome contacts. Can the authors quantitative image Ca^{++} content in mitochondria-lysosomes with positive SPLICS signal vs with negative SPLICS signal? Can they also measure quantity of TFEB on membranes of mitochondria-lysosomes with contacts (SPLICS positive) vs no contacts (SPLICS negative)?

Minor Comments

- There are language and grammatical errors throughout the manuscript. These are not major, but will require correction prior to publication.

RESPONSE TO REVIEWERS' COMMENTS

Reviewer #1 (Remarks to the Author):

In their paper "A novel SPLICS reporter reveals α -synuclein mediated regulation of lysosomes-mitochondria contact sites and TFEB nuclear translocation", Giamogante and co-workers describe a genetically encoded SPLICS reporter engineered to measure short- and long- juxtapositions between mitochondria and lysosomes. The authors show that the overexpression of α -synuclein (α -syn) strongly reduced mitochondria/lysosomes contacts and affected the transfer of Ca^{2+} from lysosomes to mitochondria, which enhanced TFEB nuclear translocation. The study is well conducted and comprehensive, however some issues reduce the enthusiasm of this reviewer.

The authors have previously used a similar approach to determine ER-mitochondria contact sites (Vallese et al., 2020 doi: 10.1038/s41467-020-19892-6, Cali and Brini, 2021 doi: 10.1007/978-1-0716-1262-0_23). However, in the 2020 paper they used live imaging of ER-mitochondria contact sites in zebrafish RB neurons to prove their premise. In this work the authors only use HeLa cells, that are not the most convincing model to study mitochondria-lysosome contact sites in polarized cells like neurons, neither to extrapolate to neuronal demise in Parkinson's disease. So, the authors should validate SPLICSS/L-P2AMT-LY reporter in neurons and if possible in vivo.

We are glad to see that the reviewer found the manuscript well conducted and comprehensive and thank she/he. We also agree with the comment therefore we have validated our new reporter in three different settings: i) mouse primary cortical neurons, Rohon-Beard sensory neurons in living Zebrafish animals and in Wing sensory neurons of living Drosophila flies. The results of these experiments in vivo and the quantitative analysis is shown in the new figure 5.

In particular, the whole section dealing with α -synuclein effect on mitophagy and the correlations made by the authors need to be revised.

The authors show that α -syn WT, A30P and A53T drastically decreased the number of short contact sites compared to Mock cells, which is intriguing, since CCCP increased lysosome-mitochondria contacts to trigger mitophagy. Several reports already shown that α -synuclein binds to mitochondrial membranes and induces its dysfunction and fragmentation, similar to CCCP. Moreover, Choi and colleagues (doi.org/10.1038/s41593-022-01140-3) showed that α -synuclein converts from a monomeric state into two distinct oligomeric states in neurons in a concentration-dependent and sequence-specific manner. This intracellular seeding events occur preferentially at mitochondrial membranes, leading to the impair complex I activity and increase mitochondrial reactive oxygen species (ROS) generation, which accelerates the oligomerization of A53T α -Syn and causes permeabilization of mitochondrial membranes and cell death. Nevertheless, authors say that an increase in α -Syn expression levels results in the activation of TFEB, possibly as a consequence of its action on lysosomes-mitochondria contact sites that leads to reduced Ca^{2+} buffering by mitochondria. Would this lead to an increase of the expression of lysosomal enzymes and LC3, so an increase in autophagy/mitophagy? However, the authors in HeLa cells expressing α -syn WT under HBSS treatment show no activation of macroautophagy!

We appreciated the comment on possible action of α -synuclein on mitochondrial function, mitophagy and mitochondria/lysosome contact sites in respect to its monomeric or oligomeric states. However, the main aim of the manuscript is not to study these correlations. The section entitled "Modulation of lysosomes-mitochondria contact sites during autophagy and mitophagy" is not dealing with α -synuclein effect on mitophagy but on the validation of the new SPLICS probe for

mitochondria-lysosome contact sites under “pharmacological” conditions well known to impact on mitochondria and lysosome’s function, i.e., to induce auto/mitophagy and mitochondria fusion with lysosomes. Rather, we were interested in studying the effects of α -synuclein on lyso mito contact sites under conditions that lead to a diffuse cytosolic distribution of α -synuclein (see immunocytochemistry) and do not impact on mitochondrial function or autophagy. HeLa cells transiently transfected with α -synuclein do not display enhanced autophagy (as documented by Supplementary Figure 5c showing no increase in GFP-LC3 puncta formation upon α -synuclein overexpression) nor mitochondria morphology alterations (see TOM20 staining), as previously documented (see 10.1074/jbc.M111.302794). We are in an early phase where α -synuclein induces a “protective” TFEB nuclear translocation. It is too early to see transcriptional effect of TFEB and the level of expression of α -synuclein is not sufficient to induce impairments (as also suggested in the next point). It will be very interesting to study a possible α -synuclein dose-dependent effect on LY-MT contacts as we have previously done for ER-MT (see also 10.1074/jbc.M111.302794 and 10.3390/cells8091072) but these aspects are not the purpose of the present study. Interestingly, we have found that upon HBSS treatment, α -synuclein overexpressing cells failed to respond with GFP-LC3 dots formation and still have reduced Ly-Mt contact sites. These results are not completely clear but, since HBSS does not impact on Ly- Mt contact sites formation (both in control and α -synuclein overexpressing cells, see Figure 3 and Supplementary Figure 5), we can speculate that under conditions that reduce Ly-Mt contacts, the cells failed to respond with enhanced autophagy to HBSS starvation. Obviously, other mechanisms could be involved, but their investigation is out of scopus of the present paper. We have decided to move this figure to the supplementary Material section.

Indeed, in 2013, Decressac and coworkers (doi: 10.1073/pnas.1305623110) showed a bi-temporal response of α -synuclein overexpression regarding the subcellular localization of the TFEB. They found that an early response to α -syn overexpression appeared to be associated with TFEB translocation into the nucleus, whereas at a later stage, deficiency in autophagy was linked to a reduced expression in the nuclear fraction due to the fact that α -synuclein binds to TFEB contributing to sequestration of this transcription factor into the cytoplasm, thus hampering the ALP response and, thereby, its own clearance. This data should be discussed in light of the authors new results. Additionally, the role of mitochondria-lysosome contact sites in mitophagy in polarized cells, like neurons, is probably different than in HeLa cells. This should be at least discussed. The paper has merit, although major revisions are required.

This is exactly the point we wanted to make. We have discussed and cited the above-mentioned manuscript in the previous version of our paper and apologize whether we were not clear enough. In our experimental conditions, i.e. 24 hours after transfection, we have found enhanced nuclear TFEB translocation in a synuclein overexpressing cells but not autophagy induction or impairment, as shown by the comparison between the black histograms on Supplementary Figure 5c. In agreement with Decressac and coworkers, we do believe that enhanced TFEB nuclear translocation is an early protective response to α -synuclein overexpression. In addition, we here propose that this response is triggered by α -synuclein action on mitochondria-lysosome contact sites and reduced Ca^{2+} buffering by mitochondria. We have now better discussed these results and the cited manuscript in the “ α -Synuclein was involved in TFEB cellular localization “. We have also performed a new experiment shown in the new Figure 8, showing that the α -synuclein induced TFEB nuclear translocation is indeed dependent on the Ca^{2+} microdomain since it could be fully inhibited by treatment with the Ca^{2+} chelator BAPTA-AM.

Reviewer #2 (Remarks to the Author):

Summary.

In this manuscript, Giamogante and colleagues designed and described, in a well-written manuscript, a couple of new probes to visualize short and long lysosome-mitochondria contacts with conventional confocal microscopy. These probes were used to evaluate the impact of Rab7A GTPase and its GTPase-activating protein (TBC1D15) on the contacts between both organelles. In the same way, how these contacts were affected by different stressors and pathological conditions was studied with these probes. Once they observed this dynamic, calcium flux was evaluated and related to the contact differences with repercussions on TFEB localization. These experiments demonstrated that the probes designed by this group represent a powerful tool to evaluate the dynamic of lysosome-mitochondria contacts. However, there are several concerns that should be addressed.

Major points.

During the whole manuscript, the authors used transfection as DNA delivery and expression methodology for the probe, reporter, siRNA, and mutants. Since it is not clear how efficient transfection and co-transfection are in the experiments, the authors should explain the advantages of using transfection instead of generating stable cell lines with infection. In this way, there will be near to 100% cells with the probes, more co-expression with reporters and mutants would be reached, and the knockdown experiments would be more reliable.

We use transiently transfection since we want to be sure that our observations were not influenced by possible clonal adaptation, especially for the condition of α -synuclein overexpression. All the type of analysis, the monitoring of contact sites and calcium measurements, are easily and well performed only on positive transfected cells co-expressing the probe of interest and α -synuclein in mild conditions, i.e., at levels that do not induce its aggregation or redistribution. This is also important to rule out potential transcriptional compensation. Considering the importance of contact sites dynamicity for cell homeostasis, we do also believe that transient transfections will fully preserve contacts and organelle's physiology.

In "Generation and characterization of SPLICSS/L-P2ALY-MT reporter for lysosomes mitochondria contact sites". The authors showed the design of the construct they used and observed that the signal generated by the reporter co-localized with lysosomes and mitochondria. To discard that this signal is a false positive, it is suggested to use both protein reporters separately. In the same way, they should mention whether this reporter works if the big part of GFP is linked to the mitochondrial protein and the β 11 strand to the lysosomal protein. The authors showed that the mitochondrial protein reporter 100% localizes with mitochondrial markers, but the same is not observed with the lysosomal protein reporter. The authors are advised to use the same anti-GFP antibody that was used to confirm the mitochondrial part.

All the experiments performed in the paper were done by using the the SPLICS-P2A^{LY-MT} reporter which guarantees the equimolar expression of OMM-targeted GFP1-10 and a lysosomal targeted β Strand-11; both protein reporters used separately are currently employed to validate the targeting and the proper complementation process.

We understood the point raised by the reviewer and performed the requested experiments. The new images have been added to panel f of Figure 1. The validation of lysosomal targeting has been evaluated by two different approaches:

- 1) by co-expressing the lysosomes targeted GFP₁₋₁₀ (Ly-GFP₁₋₁₀) with a cytosolic S-11 tagged Kate (Kate-β11) and testing the co localization of the complemented green fluorescence signal with LAMP1 (in panel f, upper images).*
- 2) by expressing the Ly-GFP₁₋₁₀ alone, revealing its localization by the anti GFP antibody, as suggested by the reviewer, and testing the colocalization of the GFP signal with that of LAMP1 (in panel f, lower images).*

In both the cases we observed the full colocalization of lysosomal reporter with the anti LAMP1 signal.

In “Modulation of lysosomes-mitochondria contact sites during autophagy and mitophagy”. The authors should work on finding the biological reasons why there is an increase in the contact number upon the stressors. What would happen if there were no more contacts? Is this increase in contact number Rab7A-dependent? These questions might be answered using the Rab7A T22N mutant. What happens if the lysosome activity is blocked with bafilomycin or chloroquine? More or less contacts? The authors are suggested to evaluate the contact number under these conditions. Are these contacts favoring mitophagy? A mitophagy reporter might be used to evaluate this upon CCCP or OA/AA treatment and with Rab7A mutants. Finally, the authors should explain the differences between HBSS and rapamycin treatment, both inhibit mTOR and increase autophagy but only rapamycin increases short contacts.

This is a very interesting point and of course matter of future and intense research. These experiments are important to define the molecular aspects of the mitochondria-lysosome interface under physiological conditions and upon activation/inhibition of specific cellular routes, this would require a project per se. In the present paper we were not interested in studying why lysosomes-mitochondria contact sites changes upon mitophagy and autophagy, but rather to develop a tool able to monitor contact sites under physiological conditions. The aim of our experiments is to test the flexibility of the reporter to detect changes in a given interface when genetically manipulating it in a direction or another. At the moment, we don't know whether the contacts are favoring mitophagy or vice versa, we and others have shown that there is a crosstalk between these two organelles under basal conditions which is important for the normal physiology of the cell and that the distance plays a role in different physiological contexts. As for the Rapamycin and the HBSS treatment, we do believe that the nutrient sensing role of lysosomes might play a role: in terms of nutrients the two treatments are different and the consequences on mitochondria function and morphology could be also different. mTOR inhibition is of course a common outcome of HBSS and rapamycin treatment and play a key role in mitochondria function and shape as shown in 10.1038/ncb2220, but starvation or different mTOR inhibitors could differently impact on mitochondrial elongation as well as lysosomal exocytosis or other processes (as shown in 10.1073/pnas.1107402108 and 10.1016/j.molcel.2017.08.013) that could explain the differences between HBSS and rapamycin treatment, despite both inhibiting mTOR and increasing autophagy .

It is recommended to repeat the WB of Figure 5e to have a better representative image, more equal LAMP1 detection (big differences in the complete blot image are shown), and better quantification. In case the WB is not successful, the use of another lysosomal marker is recommended.

Thank you for the comment, we have replaced the WB of the original Figure 5e (now Figure 6e) with a better one.

Since it is shown that extracellular calcium affects mitochondrial calcium uptake in Figure 5i, it is recommended to evaluate the participation of extracellular calcium in the increased mitochondrial calcium uptake modulated by Rab7A Q67L and TBC1D15 D39A or its knockdown.

Thank you for the observation, we want to point out that the two experimental settings are different for a specific reason. In the case of Rab7A and TBC1D15 genetic manipulation we wanted to assess whether increase or decrease in LY-Mt contact sites could impact on mitochondrial Ca²⁺ uptake upon its release from lysosomes. We have decided to perform these measurements in the presence of extracellular calcium, mostly reflecting the physiological conditions and assuming that the impact of Ca²⁺ influx could be the same in the different conditions. In the experiment performed in α -synuclein expressing cells, we wanted to avoid any potential α -synuclein dependent contribution on cellular Ca²⁺-routes since α -synuclein has been previously reported to influence Ca²⁺ influx from the extracellular ambient (see refs 10.1242/jcs.180737; 10.1111/jnc.13045; 10.1523/JNEUROSCI.2617-07.2007; 10.3389/fnmol.2019.00237). For this reason, we decided to perform mitochondria Ca²⁺ measurements in the absence of extracellular Ca²⁺.

In “ α -Synuclein overexpression counteracts the increase of Ly-Mt short interactions induced by lysosomal cholesterol accumulation”. The authors should prove evidence that lysosomal cholesterol is still increased with U18666A treatment and α -syn WT expression.

We thank the reviewer for the comment, we have performed the experiment suggested and added it in a new Supplementary figure 4.

In “ α -Synuclein overexpression counteracts Ly-Mt short interaction under starvation”. There is no reason to use HBSS as a stressor here since in Figure 3 the authors showed that this stress does not affect the lysosome-mitochondria interactions. On the other hand, rapamycin, CCCP or OA/AA are better stimuli since both autophagy and Ly-Mt short interactions are increased.

We apologize for the misunderstanding, we should stress that this condition was selected essentially for that reason, the rationale in using HBSS is exactly that to evaluate whether in conditions that per se do not affect Ly-mt contact sites (but induce starvation) α -synuclein could impact on the contacts. But, thanks to the reviewer comment, we understood that the original title of the section could be misleading, and these aspects deserve more investigations. Therefore, we have moved this figure to the supplementary information section in favor of the in vivo experiments to strengthen more on the versatility of the SPLICS probe and its technological possible applications.

In “ α -Synuclein was involved in TFEB cellular localization”. Western blot experiments are recommended to evaluate the phosphorylation of TFEB (anti-phospho TFEB) and other mTOR targets (S6K, S6 or 4E-BP1) to confirm that the mTOR-TFEB branch is inhibited but the canonical mTOR branch is not affected. The activity of TFEB is recommended to evaluate by qPCR of TFEB target genes. Since the authors suggest that the nuclear translocation of TFEB when α -syn is overexpressed is due to an increase in calcium localized in microdomains and calcineurin-dependent de-phosphorylation of TFEB, it is suggested to evaluate TFEB phosphorylation by WB and localization by IF or using GFP-TFEB construct when Calcineurin is knockdown and α -syn overexpressed.

Nuclear translocation is a direct and fast readout of TFEB activation and can clearly be detected. In our early conditions of α -synuclein overexpression and transient transfection we do not expect transcriptional responses therefore we believe that the evaluation by qPCR of TFEB target genes would not be informative from the point of view of the contact between mitochondria and lysosomes, which is a very upstream event. To unequivocally determine whether TFEB subcellular localization was affected by α -synuclein overexpression via a calcium-dependent mechanism, we treated α -synuclein -overexpressing cells with BAPTA-AM, a well-known calcium chelator. Notably, as shown in Figure 8 e,f, we found that BAPTA treatment completely rescued TFEB nuclear localization in α -synuclein expressing cells. These data further support that calcium plays a central role in promoting TFEB nuclear translocation upon α -synuclein expression.

It is important to note that calcium is known to affect TFEB subcellular localization by multiple mechanisms, involving either activation of calcineurin or TRPML1-dependent inhibition of RagC/D GTPases. We agree with the reviewer that more work is needed to establish how mechanistically TFEB activity is affected by α -synuclein-mediated action on calcium microdomains at the lysosomes-mitochondria interface. However, this implies a considerable amount of work which we believe is out of the scope of the current manuscript and will be the subject of future studies.

Minor points.

Line 668 “t” to “T” after the period.

Fixed, thank you.

Reviewer #3 (Remarks to the Author):

In this manuscript, Giamogante et al describe and validate the development of a new probe that functions as a distance sensor between the membranes of mitochondria and lysosomes, and a reporter of contact between these two organelles in response to molecular manipulations and metabolic and proteostatic stresses. The experiments are well executed with appropriate controls and statistical analyses, and the data in general support the conclusions.

Fundamentally, the manuscript is centered on the development and validation of the SPLICS reporter probe. Results from the latter are used for further experimentation on the effects of α -synuclein uclein on lysosome-mitochondria crosstalk, and lysosome-based function and signaling. This part of the manuscript is not very strong in terms of a conceptual advance, and especially for titling the manuscript after its findings. Technically, it is not accurate to claim that the SPLICS reporter reveals α -synuclein uclein mediated regulation of TFEB nuclear translocation. The paper would be much stronger as a technical paper that reports and validates a new probe rather than over-extending incremental findings on α -synuclein uclein, and making them a central theme of the paper. The α -synuclein uclein data are also subject to the caveat that were conducted in HeLa cells, which is not a physiologically relevant cell system for studies of α -synuclein uclein-triggered pathology. Moreover, the α -synuclein uclein results seem to reproduce similar results reported by others in other systems. Therefore, they are confirmatory in nature and low in novelty.

We thank the reviewer for the comments and for considering that the experiments are well executed with appropriate controls and statistical analyses. To stress more on the technical part, we have now added a new in vivo analysis of the reporter. See new figure 5 where validation in three different settings is shown: i) mouse primary cortical neurons, Rohon-Beard sensory neurons in living Zebrafish animals and in Wing sensory neurons of living Drosophila flies. We agree with the reviewer that the data obtained in HeLa cells overexpressing α -synuclein are not relevant for studies of α -synuclein-triggered pathology, indeed we do not claim to do this. We have just taken into advantage the use of a simple cell model that we have previously characterized in terms of organelles contact sites remodelling in response to increase of α -synuclein expression (see 10.1074/jbc.M111.302794 and 10.3390/cells8091072). The novelty here, in respect to previous study by Decressac and co-workers is that we correlated the α -synuclein effect on TFEB nuclear translocation with the modulation of lysosomes-mitochondria contact sites and Ca^{2+} signals and suggested this could be an early protective response to cope against enhanced α -synuclein abundance. Furthermore, now we also show that our new SPLICS probe is suitable to be used in vivo both in Drosophila and zebrafish thus open the way to study α -synuclein -triggered pathology in more appropriate models. To give more relevance to the in vivo study we have decided to move the results of the original Figure 6 to the supplementary figures.

We agree on the comment on the title. We have now changed it in: A novel SPLICS reporter reveals α -synuclein mediated regulation of lysosomes-mitochondria contact sites which impinges on TFEB nuclear translocation.

Major Concerns:

- The short and long linkers are designed for reporting interactions between 4 and 10 nm distances. However, it is unclear whether these are bona fide organelle-organelle contacts. The manuscript

does not provide any higher- and super-resolution images of juxtaposed mitochondria and lysosomal membranes at domains of the SPLICS signal.

The authors should show higher magnification zoomed-in images of lysosome and mitochondria membranes at sites of SPLICS signal (4 and 10 nm) – ideally obtained with super-resolution microscopy – for unequivocal demonstration of a correlation between the SPLICS signal and membrane-membrane contacts.

We have modified Figure 1 by adding the zoomed inset of lysosome and mitochondria membranes at sites of SPLICS signal for 4 and 10 nm SPLICS, as suggested.

Unfortunately, there are no super resolution approaches available to go down to the resolution of 5-10nm therefore we believe they might not be completely informative for this type of contacts since it does not allow to reach high definition at 4-10 nm.

The characterization of the lysosome-mitochondria contact itself has been thoroughly and elegantly performed by the group of Kranic and others (10.1038/nature25486; 10.1073/pnas.200323611; 10.1038/s41467-021-22113-3; 10.1016/j.tins.2022.01.005; 10.1038/s41467-022-31970-5) who monitored by EM analysis the frequency of distances range from 0-5 to 15nm with a maximum peak at 5-10 nm.

The fact that with our probe we were able to detect different numbers for short and long linkers strongly argue in favor of two different and physiologically separate types of contacts, in line with we have previously shown for contact sites between ER and mitochondria. Short distances are mandatory to support the exchange of ions (such as Ca^{2+}) and or lipids, but in our experience not all contacts rely on distance, for example this is not the case for the mitochondria-peroxisomes or the ER-peroxisome interface (see also 10.1038/s41467-020-19892-6).

In the case of the 4 nm SPLICS signal, are mitochondria-lysosome contacts more extensive and tighter along the majority of the edge of each organelle in comparison to 7 nm, which are more loose and not as extensive laterally and not as closely apposed?

This is an interesting point and very difficult to assess with techniques other than EM (which would require extensive analysis and time-consuming quantification). In our experience the absolute number in the 3D rendered volume of the cell is the most reliable measure of the contact sites. Their change with distance strongly indicates that different populations of contacts are in play. The confocal microscopy does not allow to understand whether a given contact is differentially extended laterally. This is also true for the signal intensity (an extended contact and one not extended would probably be seen as one), therefore we always be more confident in quantifying the absolute number of contacts in the entire volume of the cell to state that an increase or a decrease occurred (see 10.1038/s41596-021-00614-1).

- A major concern with the Rab7 and starvation manipulations is the retrograde movement of lysosomes to perinuclear and juxtannuclear regions. In these regions, membrane density is very high, and organelle-organelle contacts might be the result of non-specific crowding. How can the authors rule out the latter possibility?

This is an interesting point as well however the intracellular distribution of the contacts in the different conditions tested including controls and conditions other than Rab7 and starvation does not appear substantially different (but an accurate evaluation would require an in vivo time-lapse analysis). In all our analysis the number of contact sites is evaluated in the 3D rendered volume of the cell and lysosomes staining was routinely performed to rule out possible changes in lysosome re-

distribution or number under the different conditions, thus ruling out non-specific crowding which would impact on the punctate and well-defined pattern of SPLICS signal and on its quantification.

The authors should spatially distinguish between SPLICS signal occurring in the perinuclear and peripheral regions, and normalize it to the surface occupied by the organelle fluorescence signal. Does SPLICS signal per organelle surface area increase in peripheral non-perinuclear regions of the cytoplasm in response to starvation and Rab7 activation?

This analysis is very complicated to perform also due to the different size of the cells and radial analysis which in our hands is very variable. On the other hand, normalizing the fluorescence with the surface in our opinion is not the best way to report changes in organelle contacts under physiological conditions since one dots of different intensity must be counted as one contact to avoid overestimation. In addition, we wanted to stress again that all our measurements were performed in the 3D rendered volume of the cell. In our experience, the absolute number of contacts per cell (since they change from plane to plane) is still the best way to quantify contacts, since the resolution of confocal microscopes does not allow to correlate the intensity with the extension of the contact. The different tethers/untethers are used here to manipulate genetically the interface providing evidence of the flexibility of the SPLICS reporter.

An advantage of the SPICS probe/technology is that it can provide spatial information on the intracellular location of organelle-organelle contacts. This aspect has not been explored in the manuscript, and seems more fitting for the manuscript than the α -synuclein experiments and directions.

The reviewer is right and of course the SPLICS signal can indeed be seen as a “marker” of the given organelle interface, in a qualitative analysis. Rather often, however the tethers that are important for a given partners do not have a punctate pattern (some of them are present in the whole organelle). This makes the analysis complicated, time consuming and not so accurate considering that the Pearson or Manders’ coefficient can only be used. In any case the newly generated probe reported here will certainly open the possibility to investigate many aspects related to lysosome-mitochondria contact sites, including functional differences in respect to their intracellular localization.

- As highlighted above, the α -synuclein experimentation is performed in HeLa cells rather than primary neurons. This is a major caveat that reduces confidence in the applicability of results in a physiological context. Moreover, the TFEB translocation in response to α -synuclein has been reported in neurons before, as the authors themselves cite. In addition, the experimental evidence is not strong enough to unequivocally demonstrate that the enhanced TFEB translocations is a result of defective Ca^{++} transfer and release from lysosomes due to defective mitochondria-lysosome contacts. More is needed to demonstrate that intra-organelle Ca^{++} concentration is directly impacted by mitochondria-lysosome contacts.

We have added an extensive characterization of the reporters in three different settings: i) mouse primary cortical neurons, Rohon–Beard sensory neurons in living Zebrafish animals and in wing sensory neurons of living Drosophila flies. The results of these experiments and the quantitative analysis is shown in the new figure 5. As mentioned above, a synuclein experiments were performed in HeLa cells since we wanted to dissect the contribution of α -synuclein in a simple cell model, independently from its pathological effect. The novelty of our results is that we have found that α -

synuclein changes the lysosome-mitochondria interface and this locally impinges on Ca²⁺ microdomains that control TFEB nuclear translocation. In support of our hypothesis, we have performed a new experiment to unequivocally determine whether TFEB subcellular localization was affected by α -synuclein overexpression via a calcium-dependent mechanism. We treated α -synuclein -overexpressing cells with BAPTA-AM, a well-known calcium chelator, and, notably, as shown in Figure 8 e,f, we have found that BAPTA treatment completely rescued TFEB subcellular localization in α -synuclein expressing cells. The results of these experiments are strikingly convincing. It is important to note that calcium release is known to affect TFEB subcellular localization by multiple mechanisms, involving either activation of calcineurin or TRPML1-dependent inhibition of RagC/D GTPases. We agree with the reviewer that more work is needed to establish how mechanistically TFEB activity is affected by α -synuclein -mediated calcium release. However, this implies a considerable amount of work which we believe is out of the scope of the current manuscript and will be the subject of future studies.

Can the authors quantitative image Ca⁺⁺ content in mitochondria-lysosomes with positive SPLICS signal vs with negative SPLICS signal? Can they also measure quantity of TFEB on membranes of mitochondria-lysosomes with contacts (SPLICS positive) vs no contacts (SPLICS negative)?

The suggested analysis is very complicated and would probably require the generation of novel proximity reporter, endowed with the ability to recognize these contacts coupled with the detection of calcium transfer. As for the quantification of TFEB on the membrane of mitochondria-lysosomes with contacts (SPLICS positive) vs no contacts (SPLICS negative) sound very difficult to experimentally investigate. However, the new experiments shown in Figure 8 strongly suggest that the effect of α -synuclein is mainly achieved at the level of the lysosomal pool of TFEB.

Minor Comments

- There are language and grammatical errors throughout the manuscript. These are not major but will require correction prior to publication.

Thank you we have checked language.

REVIEWER COMMENTS

Reviewer #1 (Remarks to the Author):

Giamogante and colleagues revised adequately their paper "A novel SPLICS reporter reveals α -synuclein mediated regulation of lysosomes-mitochondria contact sites and TFEB nuclear translocation".

Reviewer #2 (Remarks to the Author):

Overall, the authors answered all major points and performed the fastest and more relevant experiments for the paper. They left the mechanistic part for future projects. This paper is interesting but is very descriptive

Reviewer #3 (Remarks to the Author):

The authors have made an effort to address this reviewer's comments, which is appreciated. The response to the concerns regarding Rab7 enhancing the signal non-specifically due to perinuclear clustering and crowding is not satisfactory. Performing quantifications from more peripheral regions of the cell is easy to do by mask segmentation. It is not more complicated and difficult than doing it in a 3D-rendered volume. The authors argue that doing the quantifications from whole 3D cell volumes is the best way to quantify, but this analysis can result in higher contacts due to non-specific crowding of organelles in the perinuclear cytoplasm. The latter point remains a concern which has not been addressed.

In the new figure 8, the authors should show all the individual datapoints of each bar graph like they have done in all other figures. The lack of any graphs in control and BAPTA-AM conditions looks like the data is missing - understandably, this is because of very low values but the individual data points should be displayed and/or the actual mean numerical values plus/minus SEM should be shown.

RESPONSE TO REVIEWERS' COMMENTS

Reviewer #1 (Remarks to the Author):

Giamogante and colleagues revised adequately their paper "A novel SPLICS reporter reveals α -synuclein mediated regulation of lysosomes-mitochondria contact sites and TFEB nuclear translocation".

We thank the reviewer for the constructive comments that helped to improve the manuscript and we are glad to see that all the concerns have been addressed.

Reviewer #2 (Remarks to the Author):

Overall, the authors answered all major points and performed the fastest and more relevant experiments for the paper. They left the mechanistic part for future projects. This paper is interesting but is very descriptive.

We thank the reviewer, we appreciate that our efforts in answering to all major points and performing the fastest and more relevant experiments for the paper have been considered.

Reviewer #3 (Remarks to the Author):

The authors have made an effort to address this reviewer's comments, which is appreciated. The response to the concerns regarding Rab7 enhancing the signal non-specifically due to perinuclear clustering and crowding is not satisfactory. Performing quantifications from more peripheral regions of the cell is easy to do by mask segmentation. It is not more complicated and difficult than doing it in a 3D-rendered volume. The authors argue that doing the quantifications from whole 3D cell volumes is the best way to quantify, but this analysis can result in higher contacts due to non-specific crowding of organelles in the perinuclear cytoplasm. The latter point remains a concern which has not been addressed.

As suggested by the reviewer, we have performed a radial analysis from the center of the cell to the cell periphery of the SPLICS signal in control cells and in cell expressing Rab7aWT and the Rab7a mutants Q67L and T22N. The results showed no major shift of the SPLICS signal (Peaking at around 6,5 μ m), suggesting that the contacts are occurring in a similar region of the cell and the differences observed in terms of contacts number are not due to local aspecific membrane crowding. The fact that Rapamycin treatment increased the number of short mito-lysosomes contacts, but HBSS did not, and that under the same conditions the long-range contacts were instead unaffected (Figure 3) also confirms that, under our experimental conditions and treatment times, there is no direct correlation of the SPLICS with the perinuclear clustering of lysosomes (induced by both Rapamycin and HBSS). The analysis was added as new Supplementary figure 2 and the following sentence was added to the results section" Short- and long-range lysosomes-mitochondria interactions were differentially modulated by Rab7A and TBC1D15": "Radial analysis was also performed to exclude that changes in the contacts number could be due to non-specific perinuclear clustering of lysosomes and/or membrane crowding induced by Rab7 (Supplementary Fig.S2 panels a and b)".

In the new figure 8, the authors should show all the individual datapoints of each bar graph like they have done in all other figures. The lack of any graphs in control and BAPTA-AM conditions looks like the data is missing - understandably, this is because of very low values but the individual data points should be displayed and/or the actual mean numerical values plus/minus SEM should be shown.

Thank you for the comment, we have modified the histograms in Figure 8 by adding the single datapoints as suggested.

REVIEWER COMMENTS

Reviewer #3 (Remarks to the Author):

In the new radial analysis of Figure S2, the authors should draw a line indicating the position that corresponds to the line scan. If a line scan is what was performed to generate that fluorescence histogram. This was not clear from the figure legend or the response.

The quantification would be more reliable if it was performed within an expanding concentric perinuclear zone, so the signal is collectively measure from the entire perinuclear region. A line scan is subjective in terms of positioning the radius/line.

Please, clarify whether a line scan was performed or the radial analysis includes the entire perinuclear zone. It would be better if concentric regions were quantified.

RESPONSE TO REVIEWERS' COMMENTS

Reviewer #3 (Remarks to the Author):

In the new radial analysis of Figure S2, the authors should draw a line indicating the position that corresponds to the line scan. If a line scan is what was performed to generate that fluorescence histogram. This was not clear from the figure legend or the response.

The quantification would be more reliable if it was performed within an expanding concentric perinuclear zone, so the signal is collectively measure from the entire perinuclear region. A line scan is subjective in terms of positioning the radius/line.

Thank you for the comment and apologize for the misunderstanding. The radial analysis shown has been performed by using the Radial_Profile.class imageJ plugin that produces a profile plot of normalized integrated intensities around concentric circles as a function of distance from a point in the image. The point in our case is the center of each cell nucleus.

The intensity at any given distance from the point represents the sum of the pixel values around a circle. This circle has the point as its center (the center of the nucleus) and the distance from the point as radius (in our experiment 25 μ m). The integrated intensity is divided by the number of pixels in the circle that is also part of the image, yielding normalized comparable values. The profile x-axis is plotted as values according to the spatial calibration of input image. i.e., μ m distance from the center (see also [10.1242/jcs.080762](https://doi.org/10.1242/jcs.080762) for a reference).

This analysis is therefore, as suggested by the reviewer, a continuous quantification of the signal over multiple concentric areas from the center of the nucleus and radially expanding until 25 μ m of distance is reached.

We have updated the supplementary figure 2 to make this clear, and the relative figure legend. The Methods section has also been updated with the details of the reference and the type of analysis performed.

REVIEWERS' COMMENTS

Reviewer #3 (Remarks to the Author):

The authors have addressed my last comment. No further issues with the manuscript.